

**Black carbon-induced snow albedo reduction over the Tibetan Plateau: Uncertainties from**
**snow grain shape and aerosol-snow mixing state based on an updated SNICAR model**
Cenlin He[1,2], Mark G. Flanner[3], Fei Chen[2,4], Michael Barlage[2], Kuo-Nan Liou[5], Shichang Kang[6,7],
Jing Ming[8], and Yun Qian[9]
[1]Advanced Study Program, National Center for Atmospheric Research, Boulder, CO, USA
[2]Research Applications Laboratory, National Center for Atmospheric Research, Boulder, CO, USA
[3]Department of Climate and Space Sciences and Engineering, University of Michigan, Ann Arbor, MI,
USA
[4]State Key Laboratory of Severe Weather, Chinese Academy of Meteorological Sciences, Beijing, China
[5]Department of Atmospheric and Oceanic Sciences, University of California, Los Angeles, CA, USA
[6]State key laboratory of Cryospheric Science, Northwest Institute of Eco-Environment and Resources,
Chinese Academy of Sciences, Lanzhou, China
[7]CAS Center for Excellence in Tibetan Plateau Earth Sciences, Beijing, China
[8]Multiphase Chemistry Department, Max Planck Institute for Chemistry, Mainz, Germany
[9]Atmospheric Sciences and Global Change Division, Pacific Northwest National Laboratory, Richland,
WA, USA
*The National Center for Atmospheric Research is sponsored by the National Science Foundation.
*Correspondence to*: Cenlin He (cenlinhe@ucar.edu)



## Abstract

We implement a set of new parameterizations into the widely used SNow, ICe, and Aerosol Radiative (SNICAR) model to account for effects of snow grain shape (spherical versus nonspherical) and black carbon (BC)-snow mixing state (external versus internal). We find that nonspherical snow grains lead to higher pure albedo but weaker BC-induced albedo reductions relative to spherical snow grains, while BC-snow internal mixing significantly enhances albedo reductions relative to external mixing. The combination of snow nonsphericity and internal mixing suggests an important interactive effect on BC-induced albedo reduction. Comparisons with observations of clean and BC-contaminated snow albedo show that model simulations accounting for both snow nonsphericity and BC-snow internal mixing perform better than those using the common assumption of spherical snow grains and external mixing. We further apply the updated SNICAR model with comprehensive *in-situ* measurements of BC concentrations in the Tibetan Plateau snowpack to quantify the present-day (2000–2015) BC-induced snow albedo effects from a regional and seasonal perspective. The BC concentrations show distinct and substantial sub-regional and seasonal variations, with higher values in the non-monsoon season and low altitudes. As a result, the BC-induced regional mean snow albedo reductions and surface radiative effects vary by up to an order of magnitude across different sub-regions and seasons, with values of 0.7–30.7 (1.4–58.4) W m$^{-2}$ for BC externally mixed with fresh (aged) snow spheres. The BC radiative effects are further complicated by uncertainty in snow grain shape and BC-snow mixing state. BC-snow internal mixing enhances the mean albedo effects over the plateau by 30–60% relative to external mixing, while nonspherical snow grains decrease the mean albedo effects by up to 31% relative to spherical grains. Based on this study, extensive measurements and improved model characterization of snow grain shape and aerosol-snow mixing state are urgently needed in order to precisely evaluate BC-snow albedo effects.



## 1. Introduction

Snow albedo, a critical element in the Earth and climate system, can be significantly affected by light-absorbing impurities in snow (Warren and Wiscombe, 1980; Hansen and Nazarenko, 2004; Jacobson, 2004; Flanner et al., 2009; Liou et al., 2014), which further influences surface energy flux and regional hydrological cycles (Menon et al., 2010; Qian et al., 2011, 2015) through a positive snow albedo feedback (Qu and Hall, 2006). With the strongest light-absorbing ability, black carbon (BC) has been identified as one of the most important contributors to snow albedo reduction and snow melting after its deposition onto global snowpack (Ramanathan and Carmichael, 2008; Bond et al., 2013), including over the Arctic (McConnell et al., 2007; Meinander et al., 2013), North American mountains (Qian et al., 2009; Sterle et al., 2013; Skiles and Painter, 2016; Wu et al., 2018), European glaciers (Painter et al., 2013; Di Mauro et al., 2017), Asian seasonal snowpack (Wang et al., 2013, 2017; Zhao et al., 2014), and the Tibetan Plateau (Xu et al., 2009; Qian et al., 2011; Wang et al., 2015; Lee et al., 2017; Li et al., 2017, 2018; Zhang et al., 2017a, b, 2018). In addition, snow albedo can be affected by snow grain size, grain shape, and snowpack structures (Wiscombe and Warren, 1980; Flanner et al. 2007; Kokhanovsky, 2013; Liou et al., 2014; Qian et al., 2014; He et al., 2017a; Räisänen et al., 2017), which complicates the BC-snow-radiation interactions. Thus, it is critically important to account for the effects of snow grain properties and BC particles in order to accurately estimate snow albedo and subsequent hydro-climatic impacts.

The Tibetan Plateau (TP), also known as the Third Pole, is covered by the largest mass of snow and ice outside the Arctic and Antarctic (Kang et al., 2010; Yao et al., 2012). It is the source region of major Asian rivers, providing fresh water for billions of people (Qin et al., 2006; Immerzeel et al., 2010). Meanwhile, because of its thermal heating, the TP has profound dynamical influences on the atmospheric circulation in the Northern Hemisphere and long been identified to be critical in regulating the Indian and East Asian monsoons (Manabe and Terpstra, 1974; Yeh et al., 1979; Yao et al., 2012). The TP is very sensitive to the change in snow albedo and cover, which alter surface heat and water balances and further disturb the Asian hydrological cycle and monsoon climate (Kang et al., 2010). Observations have shown substantial BC concentrations in snow over the TP and suggested that BC deposition is an important driver of strong albedo reductions and accelerated glacier retreat in the region (Ming et al., 2008, 2013; Xu et al., 2009; Qu et al., 2014; Ji et al., 2015; Niu et al., 2017; Li et al., 2017b; Zhang et al., 2018). Recent studies found that BC



particles over the TP are primarily from South and East Asia, while long-range transport from northern mid-latitudinal source regions outside Asia also has nontrivial contributions (Kopacz et al. 2011; Lu et al., 2012; He et al., 2014a, b; Zhang et al., 2015; Li et al., 2016; Yang et al., 2018).

To estimate BC-induced snow albedo effects over the TP, previous studies often used observed BC concentrations in snow/ice as inputs to snow albedo models by assuming spherical snow grains and BC-snow external mixing (e.g., Ming et al., 2013; Jacobi et al., 2015; Schmale et al., 2017; Zhang et al., 2018). This simplified treatment of BC-snow interactions has been widely used in snow albedo modeling over various snow-covered regions (e.g., Warren and Wiscombe, 1980; Flanner et al., 2007; Aoki et al., 2011). However, snow grains are usually nonspherical (Dominé et al., 2003) and internally mixed with BC particles (Flanner et al., 2012) in real snowpack, which could significantly affect BC-snow albedo effects. For example, Kokhanovsky and Zege (2004) pointed out that substantial errors could occur if assuming spherical snow grains in albedo modeling. Dang et al. (2016) found that, compared with spherical snow grains, the nonspherical counterparts lead to higher pure snow albedo but smaller BC-induced albedo reduction for BC-snow external mixing. In addition, Flanner et al. (2012) showed that there could be up to 73% of BC in global snowpack internally mixed with snow grains, which increases BC-induced albedo effects by up to 86% relative to purely external mixing for spherical snow grains. Moreover, recent studies (He et al., 2014b, 2018a; Liou et al., 2014), combining both effects of snow nonsphericity and BC-snow internal mixing, revealed that the enhancement in snow albedo reduction caused by internal mixing can be weakened by snow nonsphericity effect. Therefore, ignoring these two critical factors in previous studies could lead to biased estimates of BC-induced snow albedo effects over the TP and elsewhere, which highlights the necessity of accounting for the two features together in snow albedo modeling and assessing the associated uncertainty.

In this study, we implement a set of new BC-snow parameterizations (He et al., 2017b) into the widely used SNow, ICe, and Aerosol Radiative (SNICAR) model (Flanner et al., 2007) to consider the effects of snow nonsphericity and BC-snow internal mixing. We further apply the updated SNICAR model with a comprehensive set of *in-situ* measurements of BC concentrations in the TP snowpack to estimate the present-day (2000-2015) BC-induced snow albedo effects and associated uncertainties from snow grain shape (spherical versus nonspherical) and BC-snow mixing state (external versus internal) from a regional and seasonal perspective. To the best of our knowledge, this is the first attempt to quantify BC-snow albedo effects over the TP by taking into



account the aforementioned two factors concurrently with observational constraints. We describe
BC observations in the TP snowpack in Section 2. We implement the BC-snow parameterizations
and evaluate model results in Section 3. We further quantify and discuss the BC-snow albedo
effects and associated uncertainties in Section 4. Finally, we present conclusions, implications,
and future work in Section 5.

**2. BC observations in the Tibetan snowpack**
We collect available *in-situ* observations of BC concentrations in snow/ice over the TP and
surrounding areas during 2000–2015 from historical measurements (see Table S1 for summary).
Although the features of BC concentrations at each site have been described in detail by previous
observational studies, the present analysis seeks to summarize all these measurements in order to
understand the regional and seasonal characteristics of BC pollution in the TP snowpack and more
importantly to estimate the corresponding BC-snow albedo effects and associated uncertainties
due to snow grain shape and BC-snow mixing state using an updated snow model (see Section 3).
For detailed analyses, we divide the entire TP and surrounding areas into six sub-regions
(Fig. 1), including northwestern TP (NWTP; 34–40°N, 70–78°E), north of TP (NOTP; 40–45°N,
70–95°E), northeastern TP (NETP; 34–40°N, 95–105°E), southeastern TP (SETP; 27–34°N, 95–
105°E), central TP (CTP; 30–36°N, 78–95°E), and the Himalayas (HIMA). We note that NOTP
represents the Tianshan region. Due to its proximity to the TP, we analyze it together with the TP
snowpack in this study. Moreover, BC concentrations in the TP snowpack show distinct altitudinal
and seasonal variations within each sub-region (Figs. 1a–1f), with much larger values at relatively
lower altitudes (<5200 m a.s.l.) and in the non-monsoon season (October–May; Xu et al., 2009),
compared with higher altitudes (>5200 m a.s.l.) and the monsoon season (June–September; Xu et
al., 2009), respectively. Thus, we conduct analyses according to different altitudes (above or below
5200 m a.s.l.) and seasons (monsoon or non-monsoon). In addition, for any observational site with
multiple measurements during the same season, we average the measurements to represent the
mean BC pollution condition for this site during the season. Since a rather limited number of sites
provide vertically resolved BC measurements throughout snowpack, we average BC
concentrations throughout snow layers at these sites, which may introduce some uncertainties.
Figures 1a–1f show that BC concentrations in snow are generally much higher during the
non-monsoon period than during the monsoon period by up to one order of magnitude, except for





NWTP and NOTP. This is because the four sub-regions (NETP, SETP, CTP, and HIMA) are
dominated by the strong BC emissions in the non-monsoon season (particularly winter and spring)
over South and East Asia (Lu et al., 2012; Zhang et al., 2015; Yang et al., 2018) and the efficient
wet removal of BC in Asia in the monsoon season (Xu et al., 2009; He et al., 2014a). In contrast,
the high concentrations during the monsoon period over NWTP and NOTP are primarily caused
by the enrichment of BC via sublimation and/or melting of snow (Ming et al., 2009; Yang et al.,
2015) and emissions from Central Asia and Middle East (Kopacz et al., 2011; Schmale et al.,

2017).

Furthermore, BC concentrations are consistently larger at low altitudes (<5200 m) than at

high altitudes (>5200 m) by a factor of 2–10 in each sub-region (Figs. 1a–1f), which is consistent
with previous studies (Ming et al., 2009, 2013) which suggested that BC concentrations decrease
with increasing elevations. Such altitudinal contrast in BC concentrations are maximal (with
differences larger than one order of magnitude) over HIMA and SETP. This elevational
dependence can be attributed to the stronger local emissions at lower elevations, the reduced
efficiency of BC transport to higher elevations, and the higher temperature at lower elevations
leading to stronger snow melting and hence BC enrichment in snow (e.g., Ming et al., 2013; Niu
et al., 2017; Zhang et al., 2018).

Among the six sub-regions, the high-altitude areas in HIMA and SETP show the lowest

BC concentrations (5–30 ppb) throughout the year (Figs. 1d–1f), while NETP (with only low-
altitude sites) during the non-monsoon season is most severely polluted by BC (~4300 ppb). The
results further indicate that BC concentrations in low-altitude areas across different sub-regions
are comparable (190–450 ppb) during the monsoon season but are much more variable during the
non-monsoon season (Figs. 1d–1f). The variation of BC concentrations across the sub-regions is a
result of combined effects of the aforementioned factors (e.g., regionally and seasonally dependent
impacts from BC source, transport, removal, and snow aging). We note that the current
observations over the TP are still rather limited spatially and temporally, leading to questions of
representativeness and introducing uncertainty in the analysis. Thus, the large sub-regional,
altitudinal, and seasonal heterogeneity of BC concentrations in the TP snowpack highlights an
urgent need for extensive measurements.

**3. Model description, implementation, and evaluation**



### 3.1 SNICAR model

Flanner et al. (2007) developed a multi-layer SNow, ICe, and Aerosol Radiative (SNICAR) model, which has been widely used for snowpack simulations globally. It is also coupled to global climate models (e.g., Community Earth System Model, CESM) to investigate effects of impurity contamination, snow grain properties, and snow aging on snowpack albedo. A detailed model description has been presented by Flanner et al. (2007) and implementation in CESM is described by Oleson et al. (2013). Here we briefly summarize the key model elements related to the present study. SNICAR simulates snowpack radiative transfer based on the theory from Wiscombe and Warren (1980) and the multi-layer two-stream radiative transfer solution from Toon et al. (1989). It resolves vertical distributions of snow properties, impurity distributions, and heating throughout the snowpack column, as well as impact of underlying ground surfaces. The number of snow layers can be specified by users according to research objectives. The default SNICAR model assumes spherical snow grains and external mixing between impurities and snow grains. As inputs to radiative transfer calculations, the optical properties (extinction cross section ($Q_{ext}$), single-scattering albedo ($\omega$), and asymmetry factor ($g$)) of snow grains and impurities, archived as lookup tables, are offline computed by the Mie theory based on particle size distributions and refractive indices. SNICAR utilizes clear- and cloudy-sky surface incident solar flux typical of mid-latitude winter. The input parameters for SNICAR include incident radiation type (direct/diffuse), solar zenith angle, number of snow layers with thickness, density, and grain effective radius in each layer, underlying ground albedo, and aerosol concentrations in snow. In this study, we use the stand-alone version of SNICAR (available at http://snow.engin.umich.edu/snicarcode/) and implement new parameterizations of snow nonsphericity and BC-snow internal mixing into it (see Sections 3.2 and 3.3). The updated SNICAR model is available at https://github.com/EarthSciCode/SNICARv2.

### 3.2 Implementation of nonspherical snow grains

Previous studies commonly used an effective spherical snow grain with an equal volume-to-area ratio (i.e., equal surface area-weighted mean radius; hereinafter effective radius, $R_e$) to represent its nonspherical counterpart (e.g., Fu et al., 1999; Grenfell et al., 2005). The equal-effective-radius representation works well in computing extinction efficiency and single-scattering albedo but is inaccurate for asymmetry factor (Dang et al., 2016). To explicitly resolve snow grain





219 shapes, Liou et al. (2014) have developed a stochastic snow albedo model based on a geometric-

220 optics surface-wave (GOS) approach (Liou et al., 2011; He et al., 2015, 2016; Liou and Yang,

221 2016). Further, He et al. (2017b) developed a parameterization to account for snow nonsphericity

222 effects on asymmetry factors for three typical grain shapes, including spheroid, hexagonal plate,

223 and Koch snowflake (see Fig. 1 of He et al. 2017b). They parameterized the asymmetry factor ($g_{ns}$)

224 of nonspherical snow grains as follows:

$$g_{ns} = g_{hex} \times C_g \tag{1}$$

$$C_g = a_0 \left(\frac{f_{s,x}}{f_{s,hex}}\right)^{a_1} (2R_s)^{a_2} \tag{2}$$

227 where $a_i$ ($i = 0$–2) is the wavelength-dependent coefficient available in He et al. (2017b). $R_s$ (unit:

228 μm) is equal to the snow effective radius ($R_e$) for spheroid or hexagonal plate, and $R_e$/0.544 for

229 Koch snowflake due to its complex concave shape. $f_{s,x}$ and $f_{s,hex}$ are the shape factors (i.e., ratio

230 of $R_s$ of a nonspherical grain to that of an equal-volume sphere) of a nonspherical grain ($x$:

231 spheroid, hexagonal plate, or Koch snowflake) and hexagonal plate, respectively. $C_g$ is the

232 correction factor, and $g_{hex}$ is the asymmetry factor for hexagonal shapes computed as follows (Fu,

233 2007):

$$g_{hex} = \frac{1-g'}{2\omega} + g' \tag{3}$$

$$g' = b_0 + b_1 \times AR + b_2 \times AR^2, \quad \text{for } 0.1 \le AR \le 1.0 \tag{4a}$$

$$g' = c_0 + c_1 \times \ln(AR) + c_2 \times \ln^2(AR), \quad \text{for } 1.0 < AR \le 20 \tag{4b}$$

237 where $\omega$ is the snow single-scattering albedo, and $g'$ is the asymmetry factor related to geometric

238 reflection and refraction. $b_i$ and $c_i$ ($i = 0$–2) are the wavelength-dependent coefficients available in

239 Fu (2007). $AR$ is the snow aspect ratio (i.e., ratio of grain width to length).

240   Here we implement the He et al. (2017b) parameterization (Equations 1–4) for snow

241 asymmetry factor into SNICAR to account for nonspherical shapes. Due to the coarse spectral

242 resolution (6 bands) of the parameterization, we further use a piece-wise shape-preserved

243 polynomial interpolation method (Fritsch and Carlson, 1980) to interpolate the parameterized

244 results into 470 bands (0.3–5 μm with a 10-nm resolution) used in SNICAR. The same

245 interpolation method is also applied to implementing the single-scattering co-albedo

246 parameterization for BC-contaminated snow (see Section 3.3). We use the extinction efficiency

247 and single-scattering albedo of equal-effective-radii spheres for those of the nonspherical grains.





248   Figures 2a–2c show the spectral snow asymmetry factors for different grain shapes based

249 on the updated SNICAR model. Compared with spherical snow grains, nonspherical grains

250 (particularly Koch snowflakes) result in up to ~17% smaller asymmetry factors at wavelengths <

251 ~3.0 μm, consistent with previous studies (Liou et al., 2014; Dang et al., 2016). We note that the

252 results slightly (<3%) overestimate the asymmetry factors at two spectral peaks within 1.5–2.5 μm

253 for spheroids with large sizes ($R_e \geq 500$ μm), due to parameterization uncertainties (He et al.,

254 2017b).

255   As a result of the smaller asymmetry factors, nonspherical snow grains lead to weaker

256 forward scattering and hence higher albedo relative to their spherical counterparts (Figs. 3 and S1).

257 We find up to about 2% and 27% higher albedo for Koch snowflakes in the visible (0.3–0.7 μm)

258 and near-infrared (NIR, 0.7–5 μm) bands, respectively, compared to equal-$R_e$ spheres (Figs. 3d

259 and 3e). These results show good agreement with the conclusions from previous studies (Wang et

260 al., 2017; He et al., 2018a). The results also have important implications for snow grain size

261 retrievals via the use of albedo models to match observed spectral reflectance. For example, Dang

262 et al. (2016) and He et al. (2018a) suggested that if a nonspherical grain is simulated as a sphere,

263 its effective size has to be scaled down by a factor of 1.2–2.5 to obtain the correct snow albedo.

265 **3.3 Implementation of BC-snow internal mixing**

266   Flanner et al. (2012) showed that the effect of BC-snow internal mixing can be equivalent

267 to applying an enhancement ratio to BC absorption cross sections with the external mixing

268 assumption and developed a lookup table for the enhancement ratio. Recently, He et al. (2017b)

269 explicitly resolved the structures of BC-snow internal mixtures for different snow shapes and

270 found that inclusions of BC increase snow single-scattering co-albedo (1-$\omega$) and hence absorption

271 but have negligible effects on snow asymmetry factor and extinction efficiency. They further

272 parameterized the effect of internal mixing on 1-$\omega$ as follows:

273        $$E_{1-\omega} = d_0 \times (C_{BC} + d_2)^{d_1} \tag{5}$$

274 where $E_{1-\omega}$ is the co-albedo enhancement defined as the ratio of single-scattering co-albedo for

275 BC-contaminated snow to that for pure snow, which is a function of BC mass concentration in

276 snow ($C_{BC}$, unit: ppb). $d_i$ ($i = 0–2$) is the wavelength-dependent parameterization coefficient

277 available in He et al. (2017b).



Here we implement the He et al. (2017b) parameterization (Equation 5) for snow single-
scattering co-albedo to account for BC-snow internal mixing. We note that the BC mass absorption
cross section (MAC) at 550 nm used in He et al. (2017b) is 6.8 $m^2$ $g^{-1}$ with a BC density of 1.7 g
$cm^{-3}$ and an effective radius of 0.1 μm. Thus, to obtain a BC MAC of 7.5 $m^2$ $g^{-1}$ at 550 nm
recommended by Bond and Bergstrom (2006), we adjust the BC size and density in this study. We
assume a lognormal BC size distribution with a geometric mean diameter of 0.06 μm following
Dentener et al. (2006) and Yu and Luo (2009) and a geometric standard deviation of 1.5 following
Flanner et al. (2007) and Aoki et al. (2011). Then, we tune the BC density to 1.49 g $cm^{-3}$ to match
the MAC (7.5 $m^2$ $g^{-1}$). The resulting BC size effect on $E_{1-\omega}$ is quantified using a parameterization
developed by He et al. (2018b) as follows:

$$E_{1-\omega,R_{BC}} = k_{\lambda,R_{BC}} \times E_{1-\omega,R_{BC}=0.05}^{f_{\lambda,R_{BC}}} \qquad (6a)$$

$$\text{with } d_{\lambda,R_{BC}} = \left(\frac{R_{BC}}{0.05}\right)^{m_\lambda}, \quad f_{\lambda,R_{BC}} = \left(\frac{R_{BC}}{0.05}\right)^{n_\lambda} \qquad (6b)$$

where $E_{1-\omega,R_{BC}}$ and $E_{1-\omega,R_{BC}=0.05}$ are the $E_{1-\omega}$ for a certain BC effective radius ($R_{BC}$) and a $R_{BC}$ of
0.05 μm (reference case), respectively. $k_{\lambda,R_{BC}}$ and $f_{\lambda,R_{BC}}$ are empirical parameters relying on
wavelength and BC size. $m_\lambda$ and $n_\lambda$ are wavelength-dependent coefficients available in He et al.
(2018b).
Figures 2d–2f show the spectral single-scattering co-albedo of snow internally mixed with
BC based on the updated SNICAR model. The strongest co-albedo enhancement (up to about 4
orders of magnitude for 1000 ppb BC) is in the visible band, with negligible effects at wavelengths
>1 μm. As a result of the enhanced snow absorption, snow albedo reduces about two-fold more
due to internal mixing than external mixing (Figs. 4 and S2–S4). In contrast, BC decreases snow
albedo much less for nonspherical snow grains than spherical grains (Figs. 4 and S3–S4),
suggesting an important interactive effects of snow grain shape and BC-snow mixing state on snow
albedo reductions. For example, BC-sphere external mixing leads to similar visible albedo
reductions with BC-hexagonal plate internal mixing. This is consistent with our previous findings
(He et al., 2018a). Although the internal mixing effect dominates at the NIR wavelengths (Fig. 4e),
the NIR albedo reduction is a factor of 3–5 lower than the visible reduction. Thus, both snow
nonsphericity and BC-snow internal mixing play comparably important roles in determining all-
wavelength albedo reductions (Fig. 4f). This highlights the significance of simultaneously
accounting for these two factors in accurate estimates of BC-snow albedo effects.



Moreover, to cross-validate model results, we compare the simulated snow albedo and its
reduction for BC-snow internal mixing using the He et al. (2017b) parameterization with those
using the Flanner et al. (2012) lookup table. We find very good agreement (mean differences <
3%) between the two schemes for different snow sizes and shapes (Figs. 5 and S5–S6), although
the He et al. (2017b) parameterization leads to slightly stronger and weaker albedo reductions for
higher (>1000 ppb) and lower (<1000 ppb) BC concentrations, respectively. Compared with the
lookup table method, the newly-implemented parameterization in this study can be applied to a
wider range of snow grain size, shape, and BC concentration scenarios without sacrificing
computational efficiency.

**3.4 Comparisons with observations**
**3.4.1 Pure snow albedo**
We evaluated spectral pure snow albedo from SNICAR simulations by comparing with
observations (Fig. 6) from laboratory measurements (Hadley and Kirchstetter, 2012), open-field
experiments (Brandt et al., 2011), and field measurements in the Rocky Mountains (Painter et al.,
2007) and at the South Pole (Grenfell et al., 1994). To conduct reasonable comparisons, we used
the observed snowpack conditions in model simulations (e.g., snow density, grain size, thickness,
snowpack layers, direct/diffuse light, solar zenith angle, and underlying ground albedo) and made
reasonable assumptions for cases when measurements are absent (see Table 1 and Figure 6). We
further assumed four types of snow shapes (sphere, spheroid, hexagonal plate, and Koch
snowflake) in the simulations to investigate shape effects, due to the lack of measurements.
We find that model simulations generally capture the observed patterns of spectral snow
albedo in all cases (Fig. 6). However, assuming spherical grains tends to underestimate snow
albedo in the NIR band, while using nonspherical grains improves model results. For example,
compared with the observations (Painter et al., 2007), simulations assuming snow spheres show a
systematical underestimate of up to ~0.1 at wavelengths >0.7 μm, particularly at 1.0–1.2 μm (Fig.
6c), while simulations assuming hexagonal plates well match the observations. Similarly, in the
observational case of Grenfell et al. (1994), assuming hexagonal plates and Koch snowflakes
substantially reduces model underestimates at 1.5–2.5 μm relative to assuming spheres, though
leading to a slight overestimate at 0.9–1.3 μm (Fig. 6d). In contrast, in comparison with the
laboratory measurements from Hadley and Kirchstetter (2012), the spherical assumption works





reasonably well, particularly for large sizes, with only slight (<0.05) underestimates. This is
because the snow grains created in those experiments tend to be spherical. Nevertheless, using
spheroids and hexagonal plates in this case still leads to slightly better model results for large ($R_e$
= 65 and 110 μm) and small ($R_e$ = 55 μm) grain sizes, respectively (Fig. 6a). In the observational
case of Brandt et al. (2011), they determined snow effective sizes by matching model results with
the measured NIR (1.0–1.3 μm) albedo. We find that assuming different snow shapes results in
drastically different grain sizes retrieved by matching their measured NIR albedo (Figs. 6b and
7d), with effective radii of 80 and 160 μm for spheres and Koch snowflakes, respectively. This
implies the necessity of accounting for realistic grain shapes in snow grain size retrievals.

**3.4.2 BC-contaminated snow albedo**

We further compared BC-contaminated snow albedo from SNICAR simulations with

observations (Fig. 7) from laboratory measurements (Hadley and Kirchstetter, 2012), open-field
experiments (Brandt et al., 2011; Svensson et al., 2016), and field measurements in the Arctic
(Meinander et al., 2013; Pedersen et al., 2015). Similar to Section 3.4.1, we used the observed
snowpack conditions in model simulations and made proper assumptions for cases when
measurements are absent (see Table 1 and Figure 7) to make reasonable comparisons. Due to the
lack of measurements, we assumed BC internally or externally mixed with different snow shapes
in the simulations to quantify the combined effects of BC-snow mixing state and snow grain shape.

Compared with the widely-used assumption of BC externally mixed with spherical snow

grains, we find that accounting for both internal mixing and snow nonsphericity improves model
simulations (Fig. 7). For example, assuming BC-sphere external mixing leads to a systematical
underestimate of polluted snow albedo for <2000 ppb BC compared with the observations from
Svensson et al. (2016), while assuming BC-Koch snowflake internal mixing reduces the model
underestimate (Fig. 7b), with the normalized mean bias (NMB) and root-mean-square error
(RMSE) decreasing from -0.04 to 0.01 and from 0.033 to 0.019, respectively. Similarly, in the
observational case of Pedersen et al. (2015), simulations assuming BC-spheroid external mixing
perform better than those assuming BC-sphere external mixing (Fig. 7a), reducing the NMB from
-0.012 to -0.003 and RMSE from 0.028 to 0.025. Compared with the observations of Meinander
et al. (2013), model results using spherical snow grains underestimate the spectral snow albedo
contaminated by BC (Fig. 7c), regardless of model assumptions of BC-snow mixing state. Using



nonspherical grains instead increases the simulated albedo and reduces model biases in this case,
although it is still unable to fully capture the observed pattern (Fig. 7c). Considering that snow
grains tend to be spherical in the observations from Hadley and Kirchstetter (2012), we assumed
BC-sphere external/internal mixing in the comparisons. The model results with external mixing
are systematically biased high, particularly for large BC concentrations (>110 ppb), while using
internal mixing effectively reduces the albedo overestimates (Fig. 7e). As such, the observations
fall between the results of external and internal mixing, suggesting a combination of partial
external and internal mixing would best match the observations. Compared with the way of
increasing BC MAC for BC-snow external mixing to reduce model overestimates in polluted snow
albedo, which was used in Hadley and Kirchstetter (2012), the present study provides a physically-
based alternative (i.e., internal mixing) for model improvements. In fact, it is very likely that a
large portion of BC is internally mixed with snow grains in the experiments of Hadley and
Kirchstetter (2012), since they produced the BC-contaminated snow via freezing of aqueous
hydrophilic BC suspensions.

**4. BC-snow albedo effects and uncertainties over the Tibetan Plateau**

Based on the observed BC concentrations in snow (see Section 2), we applied the updated

SNICAR model (see Section 3) to quantify the present-day (2000–2015) BC-snow albedo
reduction and associated surface radiative effects over the TP. We conducted albedo simulations
at each observational site using the measured snowpack thickness and density (see Table S1)
concurrently with BC measurements. If the snow property measurements are missing at certain
site, the data from nearby sites are used instead. We then computed the regional mean values by
averaging across all sites within each sub-region and season. We used typical effective radii of 100
μm and 1000 μm for fresh and aged snow, respectively, to demonstrate snow aging/size effects.
Due to the lack of measurements for snow grain shape and BC-snow mixing state, we considered
eight simulation scenarios with the combination of four snow shapes (sphere, spheroid, hexagonal
plate, and Koch snowflake) and two mixing states (internal and external). In the simulations, the
underlying ground albedo over the TP is 0.1 at the visible band (0.3–0.7 μm) and 0.2 at the NIR
band (0.7–5 μm), following observations (Qu et al., 2014). We adopted a solar zenith cosine of
0.65 (i.e., an angle of 49.5°), which is equivalent to the insolation-weighted solar zenith cosine in
the sunlit hemisphere. The effect of solar zenith angle on snow albedo can be approximated via



changing snow effective size (Marshall, 1989). Previous studies (e.g., Aoki et al., 2003; Dang et
al., 2016) indicated that the impact of snow shape and BC contamination decreases with an
increasing solar zenith angle. Following Dang et al. (2017), we compute all-sky snow albedo via
averages of clear- and cloudy-sky albedo weighted by cloud cover fraction. The mean cloud cover
fraction and all-sky surface downward solar radiation in different sub-regions and seasons (see
Table S2) are derived from the multi-year (2000–2015) monthly mean Modern-Era Retrospective
analysis for Research and Applications version 2 (MERRA-2) reanalysis meteorological fields
(https://gmao.gsfc.nasa.gov/reanalysis/MERRA-2/) with a spatial resolution of 0.5°×0.625°.

Figures 8a–8c show the regional mean BC-induced snow albedo reductions in different

sub-regions and seasons. The spatiotemporal distribution of albedo reductions generally follows
that of BC concentrations in snow (Figs. 1d–1f), with stronger albedo reductions in low-altitude
areas and the non-monsoon period. We find that snow albedo decreases by a factor of 2–3 more
for aged snow (Table S3) than for fresh snow (Table 2), due to larger grain sizes for aged snow.
This aging/size effect dominates the albedo reductions in most of TP sub-regions, particularly
during the monsoon season (Figs. 8a–8c). However, in severely polluted sub-regions including the
low-altitude areas of NETP, SETP, CTP, and HIMA during the non-monsoon season, the effects
of snow grain shape and BC-snow mixing state are comparable to those of snow size/aging (Tables
2 and S3). For example, BC-sphere internal mixing leads to an albedo reduction of 0.114 for fresh
snow in low-altitude CTP during the non-monsoon season, while BC-Koch snowflake external
mixing leads to a reduction of 0.119 for aged snow.

Moreover, BC-snow internal mixing enhances the mean albedo reductions by 30–60%

(relative difference) across all the sub-regions and seasons, with similar enhancements for different
snow shapes and sizes (Tables 2 and S3). For example, assuming BC-sphere external mixing leads
to an annual albedo reduction of 0.066 (0.164) for fresh (aged) snow in NETP, while the internal
mixing counterpart results in a reduction of 0.095 (0.225). Our results are partially different from
those in He et al. (2018a) which showed a stronger enhancement (relative difference) in albedo
reduction caused by internal mixing for nonspherical grains than spherical grains, due to different
environmental conditions and snow albedo models used in the two studies. We further find that
nonspherical snow grains weaken the mean albedo reductions by up to 31% relative to spherical
grains in different sub-regions and seasons, with the strongest weakening for Koch snowflakes



(Figs. 8a–8c). The nonsphericity effect is smaller for aged snow compared with fresh snow (Tables
2 and S3), consistent with our previous findings (He et al., 2018a).
Although the BC concentrations in the TP snowpack tend to dominate the regional and
seasonal pattern of snow albedo reductions for fresh/aged snow (Figs. 1d–1f and 8a–8c), the
combined effects of snow grain shape and BC-snow mixing state can complicate the picture. For
example, with the widely used assumption of BC externally mixed with snow spheres, the non-
monsoon albedo reductions are 0.034 and 0.067 for high-altitude CTP and low-altitude SETP with
BC concentrations of 332 and 1111 ppb in fresh snow, respectively. However, if BC particles were
internally mixed with snow spheres in CTP and externally mixed with Koch snowflakes in SETP,
the albedo reductions in the two areas would become the same (0.047), regardless of the
substantially different BC concentrations. This points toward an imperative need for both extensive
measurements and improved model characterization of snow grain shape and aerosol-snow mixing
state for accurate quantification of BC-induced snow albedo reductions over the TP and elsewhere
with strong heterogeneity of snowpack properties and contamination.
Figures 8d–8f show the regional mean surface radiative effects caused by BC-induced
snow albedo reductions, which vary from 0.7 to 11.2 W m$^{-2}$ across different sub-regions during
the monsoon season and from 1.2 to 30.7 W m$^{-2}$ during the non-monsoon season for BC externally
mixed with fresh snow spheres. The sub-regional variation increases to 1.4–37.7 W m$^{-2}$ and 3.5–
58.4 W m$^{-2}$ for aged snow during the monsoon and non-monsoon periods, respectively (Tables 3
and S4). In general, the spatiotemporal distribution of surface radiative effects follows that of snow
albedo reductions (Figs. 8a–8f). The impacts of snow nonsphericity and BC-snow internal mixing
on the surface radiative effects are similar to those on the albedo reductions discussed above. The
maximum surface radiative effect over the TP can reach up to 45.4 (79.9) W m$^{-2}$ in NETP during
the non-monsoon season for BC internally mixed with fresh (aged) snow spheres (Tables 3 and
S4). The mean BC-induced snow albedo effects in the relatively clean TP areas (e.g., high-altitude
HIMA and SETP) are comparable to those over the Arctic and North American snowpack (Dang
et al., 2017; He et al., 2018a), while the effects in the contaminated TP areas (e.g., low-altitude
HIMA, CTP, SETP, and NETP) are generally similar to those in the low-elevation Alps (Painter
et al., 2013) and northern China snowpack (Wang et al., 2017).
Previous studies have shown accelerated snowmelt caused by BC-snow albedo effects in
the TP. For example, Yasunari et al. (2010) estimated that BC-induced albedo reductions over





Himalayan glaciers could result in an extra snowmelt of 1–7 mm day$^{-1}$ during the melting/summer
season. Qian et al. (2011) found a BC-induced snowmelt of up to 1.3 mm day$^{-1}$ in late spring and
early summer averaged over the entire TP. Our results further suggest that the uncertainty
associated with snow shape and BC-snow mixing state could lead to a substantial variation in BC-
induced albedo reduction and hence snowmelt, which has significant implications for runoff and
water management in Asia. Accurate quantifications of the impact of snow grain shape and BC-
snow mixing state on snowmelt and subsequent hydrological processes require interactive land
surface and/or climate modeling, which will be investigated in future work.
We note that the present estimates of BC-induced snow albedo effects have uncertainties
and limitations. For example, different techniques have been used to measure BC concentration in
snow/ice, which may lead to discrepancies and inconsistency among observations and in model-
observation comparisons (Qian et al., 2015 and references therein). Besides, BC measurements
across the TP are from various sample types, such as surfaces of snowpack (with fresh/aged snow)
and glacier (with both snow/firn and granular ice), which may introduce uncertainty to the
understanding of BC contamination patterns (Zhang et al., 2017a; Li et al., 2018). In addition, in
the model, we do not account for the vertical variability of BC and snow grain properties in the TP
snowpack as well as some complex snowpack processes, including dynamic snow aging and
melting, post-depositional enrichment, and melting water scavenging, which may exert nontrivial
effects on BC-snow albedo effects (e.g., Flanner et al., 2007; Qian et al., 2014; Dang et al., 2017).
Improved estimates require comprehensive climate modeling coupled with the updated SNICAR
snow model and constrained by observed BC and snowpack conditions.

**5. Conclusions, implications, and future work**
We implemented a set of new BC-snow parameterizations into SNICAR, a widely used
snow albedo model, to account for the effects of snow nonsphericity and BC-snow internal mixing.
We evaluated model simulations by comparing with observations. We further applied the updated
SNICAR model with a comprehensive set of *in-situ* measurements of BC concentrations in the
Tibetan Plateau (TP) snowpack (glacier) to quantify the present-day BC-induced snow albedo
effects and associated uncertainties from snow grain shape and BC-snow mixing state.
We found that nonspherical snow grains tend to have higher albedos compared with
spherical snow grains, while BC-snow internal mixing leads to much larger albedo reductions





relative to external mixing. The albedo reductions are weaker for nonspherical snow grains than
spherical grains, implying an important interactive effect from snow nonsphericity and internal
mixing. These results are consistent with previous studies (Flanner et al., 2012; Dang et al., 2016;
He et al., 2018a) and highlight the importance of concurrently accounting for snow grain shape
and aerosol-snow mixing state in snow albedo and climate modeling.

Comparisons with clean snow observations showed that model simulations using spherical

snow grains generally capture the observed spectral albedo but lead to a systematic underestimate
at NIR wavelengths, while assuming nonspherical snow grains improves model results. Further
evaluation with observed BC-contaminated snow albedo indicated that model simulations with the
combined effects of snow nonsphericity and BC-snow internal mixing perform better than those
with the common assumption of BC externally mixed with snow spheres.

We collected available *in-situ* observations of BC concentrations in snow/ice over the TP

during 2000–2015. We found that BC concentrations show distinct sub-regional and seasonal
variations. The concentrations are generally higher in the non-monsoon season and low-altitudes
(<5200 m) than in the monsoon season and high-altitudes (>5200 m), respectively. Among
different sub-regions, the high-altitude areas in the Himalayas and southeastern TP show the
lowest mean BC concentrations (<30 ppb) throughout the year, while the northeastern TP during
the non-monsoon season is most severely polluted by BC (>4000 ppb). The substantial
spatiotemporal heterogeneity of BC concentrations in the TP snowpack implies an urgent need for
more extensive measurements.

Based on the observed BC concentrations and snowpack properties, we conducted

SNICAR simulations to quantify the BC-induced snow albedo reductions and associated surface
radiative effects in different sub-regions and seasons. The spatiotemporal distribution of albedo
reductions generally follows that of BC concentrations, with stronger albedo reductions in the non-
monsoon period and low-altitude areas. We found that the effects of snow grain shape and BC-
snow mixing state become comparably important with the snow aging/size effect over severely
polluted areas. BC-snow internal mixing enhances the mean snow albedo reductions by 30–60%
relative to external mixing across different sub-regions and seasons, while nonspherical snow
grains weaken the albedo reductions by up to 31% relative to spherical grains. Therefore, the
combined effects of snow grain shape and BC-snow mixing state can complicate the
spatiotemporal features of BC-snow albedo reductions over the TP.





We found that the BC-induced mean surface radiative effects can vary by up to an order of
magnitude across different sub-regions and seasons, showing a similar pattern with the snow
albedo reduction, with the effects further modified by snow grain shape and BC-snow mixing state.
The maximum effect can reach up to 45.4 (79.9) W m$^{-2}$ in the northeastern TP during the non-
monsoon season, assuming BC-sphere internal mixing for fresh (aged) snow. The results suggest
that the uncertainty associated with snow shape and BC-snow mixing state over the TP could lead
to a large variation in BC-induced snowmelt, with significant implications for hydrological
processes and water management in Asia.
In summary, this study points toward an imperative need for extensive measurements and
improved model characterization of snow grain shape and aerosol-snow mixing state in order to
accurately estimate BC-induced snow albedo effects over the TP as well as other areas with highly
heterogeneous aerosol contamination and snowpack properties. In future work, we will incorporate
the new features of the updated SNICAR model into land surface and climate models, including
CESM-Community Land Model (CLM) for global modeling and WRF-Noah-MP for regional
modeling, to account for the effects of snow grain shape and aerosol-snow mixing state and to
assess the associated uncertainties and hydrological feedbacks in global/regional climate system.

***Data availability.*** Users can access the data used and produced by this study via the supplementary
materials and the corresponding author without any restrictions. The updated SNICAR model can
be downloaded at https://github.com/EarthSciCode/SNICARv2.

***Author contributions.*** CH designed and performed the parameterization implementation and
model simulations. MF offered data and help in developing model codes. FC and MB helped refine
model experiments. SK and JM provided black carbon observations. KNL and YQ gave valuable
comments. CH prepared the manuscript and all co-authors helped improve the manuscript.

***Competing interests.*** The authors declare that they have no conflict of interest.





*Acknowledgements.* C. He thanks Wenfu Tang and Roy Rasmussen for helpful discussions. C. He
was supported by the NCAR Advanced Study Program (ASP) Fellowship. The National Center
for Atmospheric Research (NCAR) is sponsored by the National Science Foundation (NSF). The
State Key Program of the National Natural Science Foundation of China is under Grant 91537211
and NCAR Water System. The contribution of Y. Qian in this study was supported as part of the
Energy Exascale Earth System Model (E3SM) project, funded by the U.S. Department of Energy,
Office of Science, Office of Biological and Environmental Research's Earth System Modeling
program. The Pacific Northwest National Laboratory (PNNL) is operated for DOE by Battelle
Memorial Institute under contract DE-AC06-76RLO 1830.

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




**Table 1.** Parameter values used in SNICAR simulations when comparing with observed snow
albedo (see Figs. 6 and 7). The observed snowpack properties are used in each case when they are
available. Four types of snow shapes (sphere, spheroid, hexagonal plate, and Koch snowflake)
and/or two types of BC-snow mixing (internal and external) are assumed in the simulations.

| Observational cases | | Model parameters | | | | | | | |
|---|---|---|---|---|---|---|---|---|---|
| References | Type | Radiation | Solar zenith angle | Underlying ground albedo | Snow layer | Snow thickness (cm) | Snow effective radius (μm) | Snow density (kg m⁻³) | BC content (ppb) |
| Pure snow | | | | | | | | | |
| Hadley and Kirchstetter 2012 | laboratory measurement | direct | 0° | 0 | 1 | semi-infinite | 55/65/110 | 550 | |
| Brandt et al. 2011 | open-field experiment | diffuse | | 0[*] | 2 | 15 / 40 | 80/95/140/160 / 500[*] | 150 / 300 | |
| Painter et al. 2007 | field measurement | diffuse | | 0[*] | 1 | 100 | 600 | 350 | 0 |
| Grenfell et al. 1994 | field measurement | diffuse | | 0.6 | multiple layers with layer-specific properties (see reference for details) | | | | |
| BC-contaminated snow | | | | | | | | | |
| Pedersen et al. 2015 | field measurement | diffuse | | 0.2 | 1 | multiple cases with case-specific properties (see reference for details) | | 150[*] | case-specific (see reference for details) |
| Svensson et al. 2016 | open-field experiment | direct | 61.3° | 0.1 | multiple layers with layer-specific snow properties & vertically averaged BC concentrations (see reference for details) | | | | 232/489/554/ 1030/6420 |
| Meinander et al. 2013 | field measurement | direct | 55° | 0[*] | 2 | 0.5 / 9.5 | 1000 / 5000 | 350 / 350 | 87.1 |
| Brandt et al. 2011 | open-field experiment | diffuse | | 0[*] | 2 | 15 / 40 | 80/95/140/160 / 500[*] | 150 / 300 | 2250 / 20 |
| Hadley and Kirchstetter 2012 | laboratory measurement | direct | 0° | 0 | 1 | semi-infinite | 55/65/110 | 550 | 110/450/860/ 1680 |

[*]The parameters are assumed in simulations due to the lack of measurements.



**Table 2.** Regional and seasonal mean BC-induced all-sky snow albedo reductions for fresh snow
over the Tibetan Plateau during 2000–2015. See Table S3 for results of aged snow.

| Region[1] | | Season | BC mean content (ppb) | Fresh snow ($R_e$ = 100 μm) | | | | | | | |
|---|---|---|---|---|---|---|---|---|---|---|---|
| | | | | External mixing | | | | Internal mixing | | | |
| | | | | Sphere | Spheroid | Hexagonal plate | Koch snowflake | Sphere | Spheroid | Hexagonal plate | Koch snowflake |
| HIMA | high alt. low alt. | monsoon | 16.3 | 0.005 | 0.005 | 0.004 | 0.004 | 0.007 | 0.006 | 0.005 | 0.005 |
| | high alt. low alt. | non-monsoon | 29.4 1151.8 | 0.008 0.066 | 0.007 0.061 | 0.006 0.052 | 0.006 0.049 | 0.011 0.098 | 0.010 0.091 | 0.008 0.077 | 0.007 0.072 |
| | high alt. low alt. | annual | 17.5 | 0.006 | 0.005 | 0.004 | 0.004 | 0.008 | 0.007 | 0.006 | 0.005 |
| CTP | high alt. low alt. | monsoon | 63.2 446.0 | 0.014 0.047 | 0.012 0.043 | 0.010 0.036 | 0.009 0.033 | 0.018 0.065 | 0.016 0.060 | 0.014 0.050 | 0.012 0.046 |
| | high alt. low alt. | non-monsoon | 331.6 1632.9 | 0.034 0.077 | 0.031 0.071 | 0.026 0.061 | 0.023 0.057 | 0.047 0.114 | 0.043 0.106 | 0.036 0.091 | 0.033 0.084 |
| | high alt. low alt. | annual | 146.3 263.5 | 0.021 0.034 | 0.019 0.031 | 0.016 0.026 | 0.014 0.024 | 0.028 0.047 | 0.026 0.043 | 0.021 0.035 | 0.019 0.033 |
| NWTP | high alt. low alt. | monsoon | 143.6 272.2 | 0.023 0.035 | 0.021 0.032 | 0.017 0.026 | 0.016 0.024 | 0.030 0.048 | 0.027 0.044 | 0.023 0.036 | 0.021 0.033 |
| | high alt. low alt. | non-monsoon | 61.1 64.7 | 0.014 0.014 | 0.013 0.013 | 0.011 0.011 | 0.010 0.010 | 0.018 0.018 | 0.017 0.017 | 0.014 0.014 | 0.013 0.013 |
| | high alt. low alt. | annual | 87.4 191.4 | 0.016 0.028 | 0.015 0.026 | 0.012 0.021 | 0.011 0.020 | 0.022 0.038 | 0.020 0.035 | 0.016 0.029 | 0.015 0.026 |
| NETP | high alt. low alt. | monsoon | 190.9 | 0.023 | 0.021 | 0.017 | 0.016 | 0.031 | 0.029 | 0.024 | 0.022 |
| | high alt. low alt. | non-monsoon | 4323.2 | 0.151 | 0.140 | 0.118 | 0.110 | 0.223 | 0.208 | 0.178 | 0.165 |
| | high alt. low alt. | annual | 823.0 | 0.066 | 0.061 | 0.051 | 0.047 | 0.095 | 0.087 | 0.072 | 0.067 |
| SETP | high alt. low alt. | monsoon | 5.2 263.6 | 0.003 0.032 | 0.002 0.029 | 0.002 0.024 | 0.002 0.022 | 0.004 0.043 | 0.003 0.040 | 0.003 0.033 | 0.003 0.030 |
| | high alt. low alt. | non-monsoon | 13.7 1110.9 | 0.005 0.067 | 0.005 0.062 | 0.004 0.052 | 0.004 0.048 | 0.007 0.098 | 0.006 0.090 | 0.005 0.077 | 0.005 0.071 |
| | high alt. low alt. | annual | 9.0 249.4 | 0.004 0.031 | 0.004 0.028 | 0.003 0.023 | 0.003 0.021 | 0.005 0.042 | 0.005 0.039 | 0.004 0.032 | 0.004 0.029 |
| NOTP | high alt. low alt. | monsoon | 368.6 | 0.040 | 0.036 | 0.030 | 0.028 | 0.055 | 0.050 | 0.042 | 0.038 |
| | high alt. low alt. | non-monsoon | 89.1 | 0.018 | 0.016 | 0.013 | 0.012 | 0.023 | 0.021 | 0.017 | 0.016 |
| | high alt. low alt. | annual | 138.3 | 0.024 | 0.022 | 0.018 | 0.016 | 0.031 | 0.028 | 0.024 | 0.022 |

[1]Six sub-regions: Himalayas (HIMA), central Tibetan Plateau (CTP), northwestern Tibetan Plateau (NWTP),
northeastern Tibetan Plateau (NETP), southeastern Tibetan Plateau (SETP), and north of Tibetan Plateau (NOTP).
Each sub-region is further divided into high (>5200 m) and low (<5200 m) altitudes.



**Table 3.** Regional and seasonal mean BC-induced all-sky surface radiative effects (W m$^{-2}$) for
fresh snow over the Tibetan Plateau during 2000–2015. See Table S4 for results of aged snow.

| Region[1] | Season | Fresh snow ($R_e$ = 100 μm) | | | | | | | |
|---|---|---|---|---|---|---|---|---|---|
| | | External mixing | | | | Internal mixing | | | |
| | | Sphere | Spheroid | Hexagonal plate | Koch snowflake | Sphere | Spheroid | Hexagonal plate | Koch snowflake |
| **HIMA** high alt. low alt. | monsoon | 1.4 | 1.3 | 1.1 | 1.0 | 2.0 | 1.8 | 1.5 | 1.4 |
| high alt. | non-monsoon | 2.0 | 1.8 | 1.5 | 1.4 | 2.6 | 2.4 | 2.0 | 1.8 |
| low alt. | | 16.2 | 15.0 | 12.8 | 11.9 | 23.9 | 22.2 | 19.0 | 17.6 |
| high alt. low alt. | annual | 1.5 | 1.4 | 1.2 | 1.1 | 2.0 | 1.8 | 1.5 | 1.4 |
| **CTP** high alt. | monsoon | 4.2 | 3.9 | 3.2 | 3.0 | 5.6 | 5.1 | 4.2 | 3.9 |
| low alt. | | 14.7 | 13.5 | 11.2 | 10.3 | 20.5 | 18.7 | 15.6 | 14.3 |
| high alt. | non-monsoon | 7.7 | 7.0 | 5.8 | 5.3 | 10.6 | 9.7 | 8.1 | 7.4 |
| low alt. | | 17.3 | 16.2 | 13.8 | 12.8 | 25.8 | 24.1 | 20.6 | 19.1 |
| high alt. | annual | 5.3 | 4.9 | 4.0 | 3.7 | 7.2 | 6.6 | 5.4 | 5.0 |
| low alt. | | 8.7 | 8.0 | 6.6 | 6.1 | 11.9 | 10.9 | 9.1 | 8.3 |
| **NWTP** high alt. | monsoon | 7.2 | 6.6 | 5.5 | 5.0 | 9.7 | 8.8 | 7.3 | 6.7 |
| low alt. | | 11.3 | 10.3 | 8.5 | 7.8 | 15.3 | 14.0 | 11.6 | 10.7 |
| high alt. | non-monsoon | 2.7 | 2.5 | 2.1 | 1.9 | 3.6 | 3.2 | 2.7 | 2.4 |
| low alt. | | 2.7 | 2.5 | 2.1 | 1.9 | 3.6 | 3.3 | 2.7 | 2.5 |
| high alt. | annual | 3.9 | 3.5 | 2.9 | 2.7 | 5.1 | 4.7 | 3.9 | 3.5 |
| low alt. | | 6.6 | 6.1 | 5.0 | 4.6 | 9.0 | 8.2 | 6.8 | 6.2 |
| **NETP** high alt. low alt. | monsoon | 6.5 | 6.0 | 5.0 | 4.6 | 8.9 | 8.2 | 6.8 | 6.3 |
| high alt. low alt. | non-monsoon | 30.7 | 28.4 | 24.1 | 22.3 | 45.4 | 42.3 | 36.2 | 33.6 |
| high alt. low alt. | annual | 15.3 | 14.0 | 11.7 | 10.7 | 21.8 | 20.0 | 16.7 | 15.4 |
| **SETP** high alt. | monsoon | 0.7 | 0.6 | 0.5 | 0.5 | 1.0 | 0.9 | 0.8 | 0.7 |
| low alt. | | 8.5 | 7.7 | 6.4 | 5.9 | 11.5 | 10.6 | 8.8 | 8.0 |
| high alt. | non-monsoon | 1.2 | 1.1 | 0.9 | 0.8 | 1.5 | 1.4 | 1.1 | 1.0 |
| low alt. | | 14.5 | 13.4 | 11.4 | 10.5 | 21.3 | 19.7 | 16.6 | 15.4 |
| high alt. | annual | 0.9 | 0.8 | 0.7 | 0.6 | 1.3 | 1.2 | 1.0 | 0.9 |
| low alt. | | 7.3 | 6.6 | 5.5 | 5.0 | 9.9 | 9.0 | 7.5 | 6.9 |
| **NOTP** high alt. low alt. | monsoon | 11.2 | 10.2 | 8.4 | 7.7 | 15.4 | 14.1 | 11.7 | 10.7 |
| high alt. low alt. | non-monsoon | 2.9 | 2.7 | 2.2 | 2.0 | 3.8 | 3.5 | 2.9 | 2.6 |
| high alt. low alt. | annual | 4.8 | 4.4 | 3.6 | 3.3 | 6.4 | 5.8 | 4.8 | 4.4 |

[1]Six sub-regions: Himalayas (HIMA), central Tibetan Plateau (CTP), northwestern Tibetan Plateau (NWTP),
northeastern Tibetan Plateau (NETP), southeastern Tibetan Plateau (SETP), and north of Tibetan Plateau (NOTP).
Each sub-region is further divided into high (>5200 m) and low (<5200 m) altitudes.





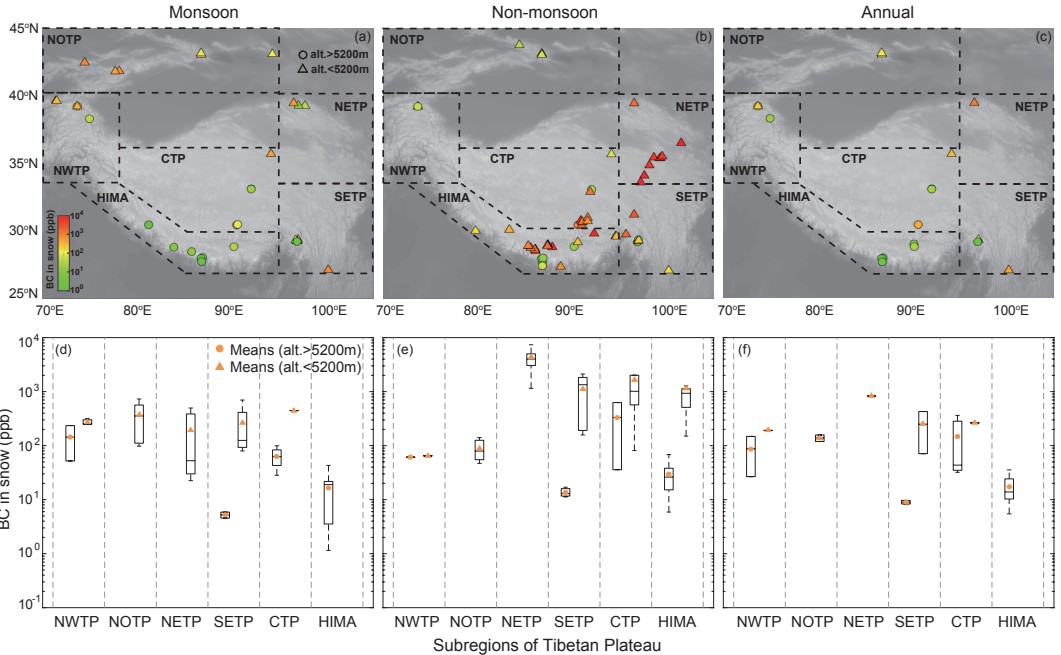

**Figure 1.** Observed BC concentrations in snow over the Tibetan Plateau (TP) during (a, d) monsoon, (b, e) non-monsoon, and (c, f) annual periods in 2000–2015 (see Table S1 for details). (a–c) Spatial distributions of seasonal mean BC concentrations at altitudes >5200 m (circles) and <5200 m (triangles) in six sub-regions, including northwestern TP (NWTP), north of TP (NOTP), northeastern TP (NETP), southeastern TP (SETP), central TP (CTP), and Himalayas (HIMA). (d–f) Boxplots of observed BC concentrations in snow (shown in a–c) within each sub-region, with medians (middle bars), interquartile ranges (between 25th and 75th percentiles; boxes), and maxima/minima (whiskers) within ±1.5 × interquartile ranges. Some boxplots are shrunk due to limited samples. Results for altitudes >5200 m and <5200 m are shown as left and right boxplots within each sub-region, respectively, with circles and triangles indicating mean values. Note that some sub-regions only have observations at altitudes >5200 m or <5200 m.






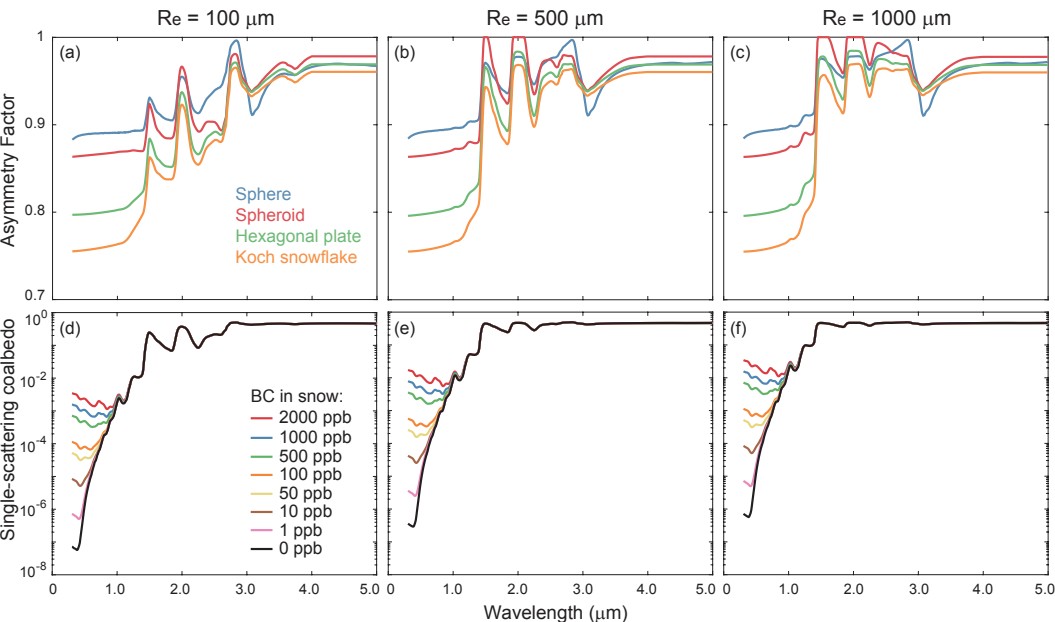

**Figure 2.** (a–c) Spectral (0.3–5 μm) asymmetry factors of pure snow with effective radii ($R_e$) of
(a) 100, (b) 500, and (c) 1000 μm for sphere (blue), spheroid (red), hexagonal plate (green), and
Koch snowflake (orange) derived from the updated SNICAR model. (d–f) Spectral single-
scattering coalbedo of snow grains internally mixed with different BC concentrations (indicated
by different colors) for snow effective radii ($R_e$) of (d) 100, (e) 500, and (f) 1000 μm derived from
the updated SNICAR model. Note that the y-axes in (d–f) are in logarithmic scales.





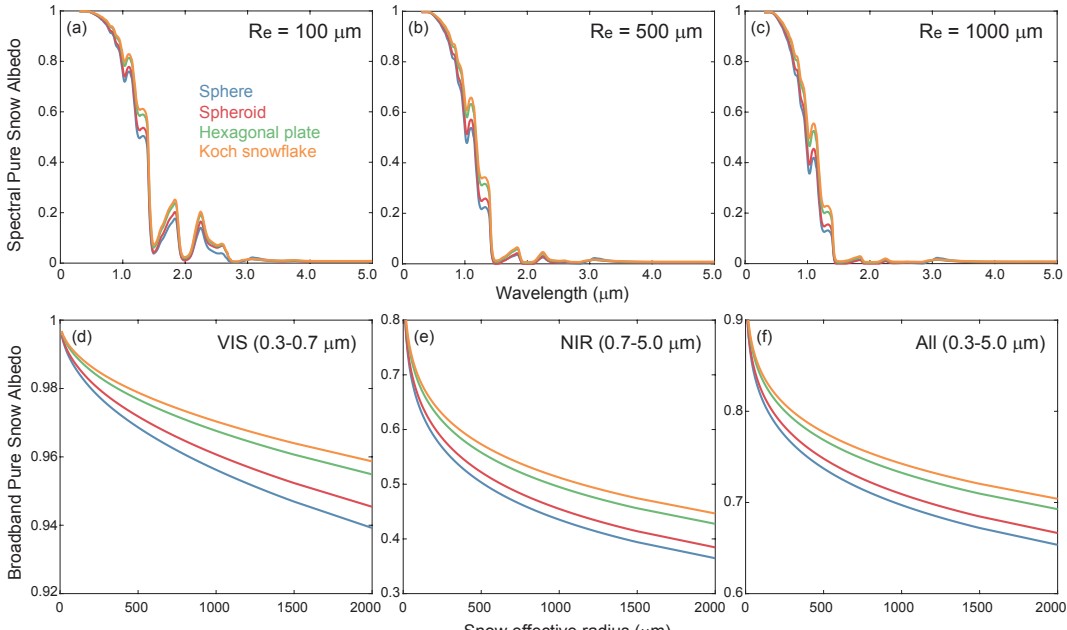

**Figure 3.** (a–c) Spectral (0.3–5 μm) direct-beam albedo of pure semi-infinite snow layers with
effective radii ($R_e$) of (a) 100, (b) 500, and (c) 1000 μm for sphere (blue), spheroid (red), hexagonal
plate (green), and Koch snowflake (orange) based on the updated SNICAR model. (d–f) Same as
(a–c), but for broadband albedo as a function of snow effective radius ($R_e$) at (d) visible (VIS, 0.3–
0.7 μm), (e) near-infrared (NIR, 0.7–5 μm), and (f) all (0.3–5 μm) wavelengths. The results for
diffuse snow albedo are shown in Fig. S1.



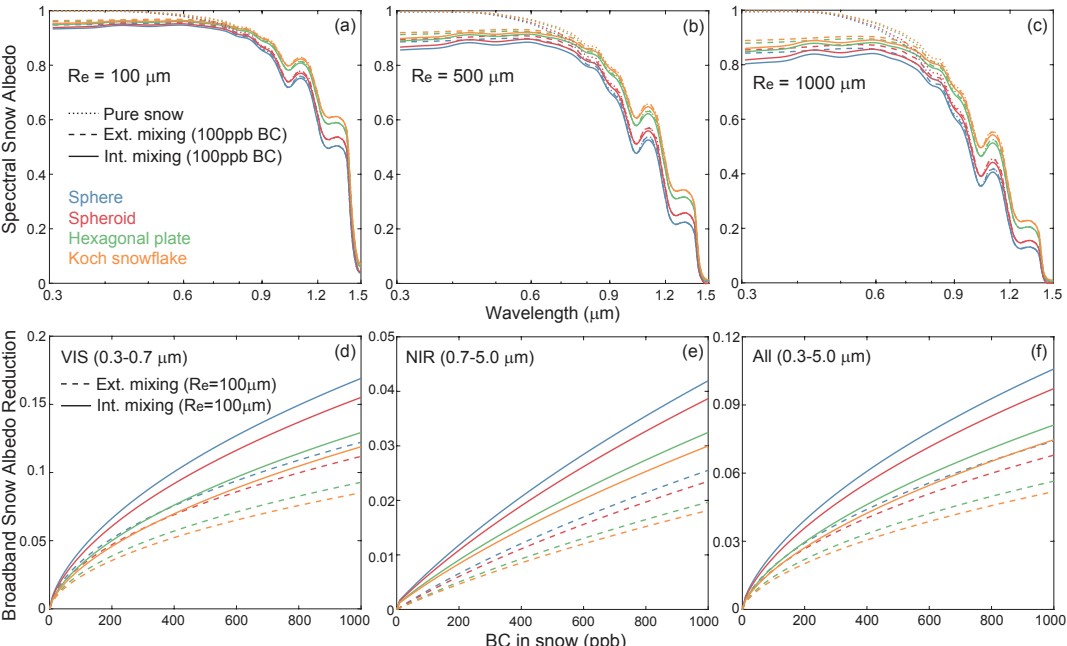

**Figure 4.** (a–c) Spectral (0.3–1.5 μm) direct-beam albedo of semi-infinite snow layers with
effective radii ($R_e$) of (a) 100, (b) 500, and (c) 1000 μm for pure snow (dotted lines), snow
externally mixed with 100 ppb BC (dashed lines), and snow internally mixed with 100 ppb BC
(solid lines) with shapes of sphere (blue), spheroid (red), hexagonal plate (green), and Koch
snowflake (orange) based on the updated SNICAR model. The results for 1000 ppb BC and diffuse
snow albedo are shown in Fig. S2. (d–f) Same as (a–c), but for broadband albedo reduction as a
function of BC concentration in snow with $R_e$ of 100 μm at (d) visible (VIS, 0.3–0.7 μm), (e) near-
infrared (NIR, 0.7–5 μm), and (f) all (0.3–5 μm) wavelengths. The results for snow with $R_e$ of 500
and 1000 μm and diffuse albedo reductions are shown in Figs. S3 and S4, respectively.





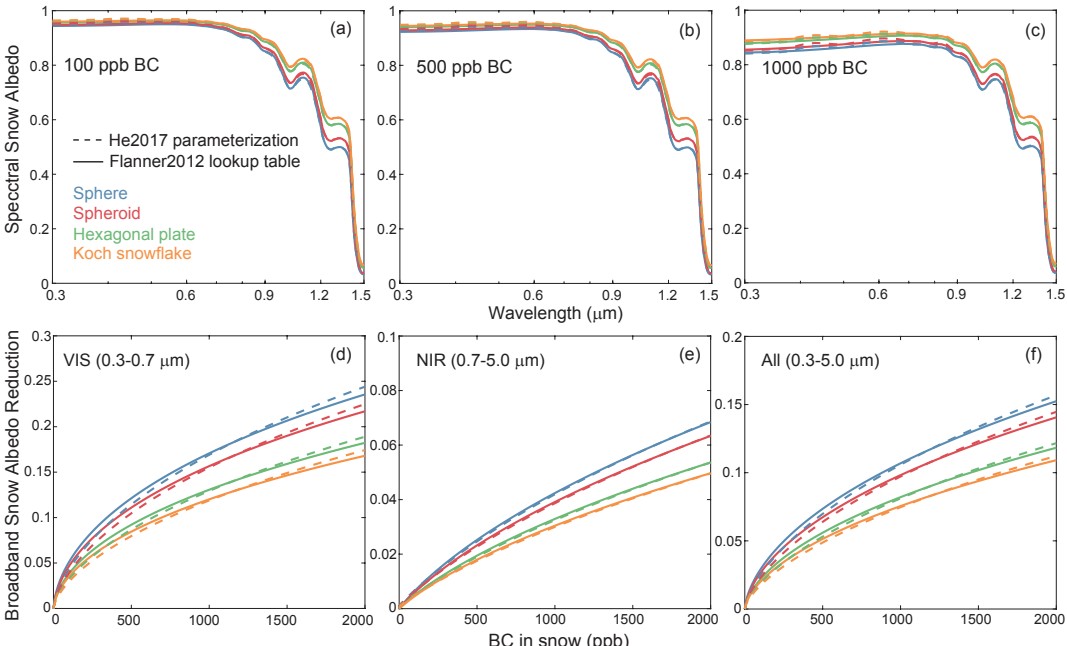

**Figure 5.** Comparisons of SNICAR simulated direct-beam albedo of semi-infinite snow layers between using the Flanner et al. (2012) lookup table (solid lines) and the He et al. (2017b) parameterization (dashed lines) for BC internally mixed with snow grains with an effective radius of 100 μm for sphere (blue), spheroid (red), hexagonal plate (green), and Koch snowflake (orange). (a–c) Spectral (0.3–1.5 μm) snow albedo for BC concentrations of (a) 100, (b) 500, and (c) 1000 ppb. (d–f) Broadband snow albedo reduction as a function of BC concentration in snow at (d) visible (VIS, 0.3–0.7 μm), (e) near-infrared (NIR, 0.7–5 μm), and (f) all (0.3–5 μm) wavelengths. The results for snow effective radii of 500 and 1000 μm are shown in Figs. S5 and S6, respectively.

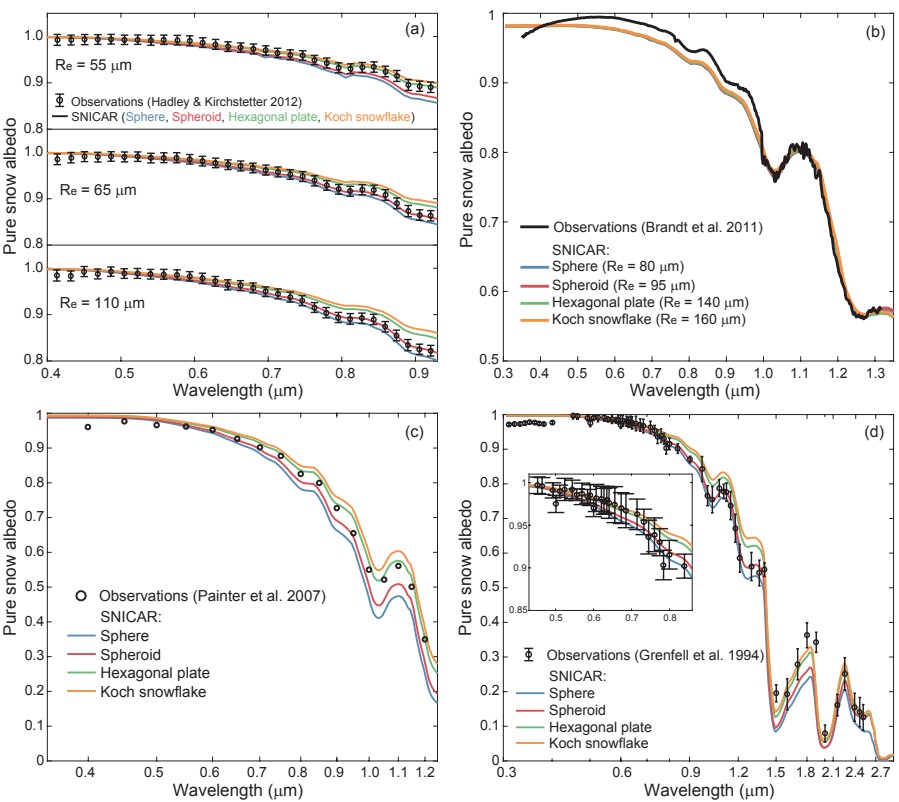

**Figure 6.** Comparisons of spectral pure snow albedo from observations (black) and SNICAR
simulations using observed snowpack properties (see Table 1) and assuming sphere (blue),
spheroid (red), hexagonal plate (green), and Koch snowflake (orange). (a) Observations are
obtained from laboratory measurements (Hadley and Kirchstetter, 2012). (b) Observations are
obtained from open-field experiments in New York (Brandt et al., 2011). The effective radii ($R_e$)
for each snow shape are obtained to best match observations at wavelengths of 1−1.3 μm. (c)
Observations are obtained from field measurements over Rocky Mountains (Painter et al., 2007).
(d) Observations are obtained from field measurements at the South Pole (Grenfell et al., 1994).



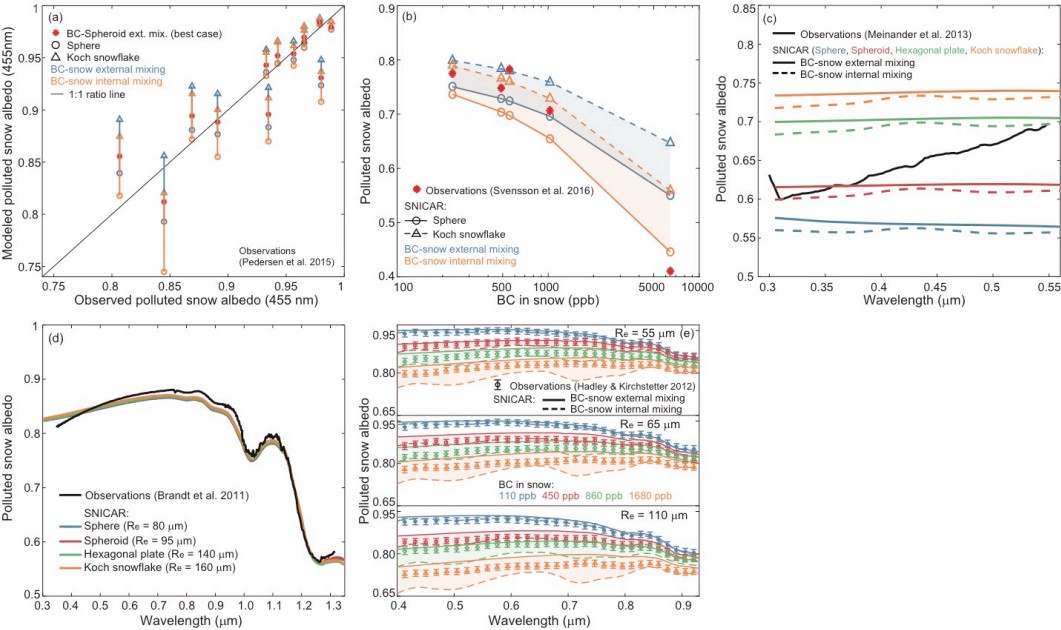

**Figure 7.** Comparisons of BC-polluted snow albedo from observations and SNICAR simulations
using observed snowpack properties (see Table 1). (a) Observations (x-axis) are obtained from
field measurements in the Arctic (Pedersen et al., 2015). Model results (y-axis) for spheres (circles)
and Koch snowflake (triangles) are shown as lower and upper limits for shape effects, along with
BC-snow external (blue) and internal (orange) mixing. Also shown is the best case (red asterisks;
BC-spheroid external mixing) that matches observations. (b) Observations (red asterisks;
broadband albedo for 0.285–2.8 μm) are obtained from open-field experiments in Finland
(Svensson et al., 2016). Model results for spheres (circles) and Koch snowflake (triangles) are
shown as lower and upper limits for shape effects, along with BC-snow external (blue) and internal
(orange) mixing. (c) Observations (black lines) are obtained from field measurements in the
European Arctic (Meinander et al., 2013). Model results assuming sphere (blue), spheroid (red),
hexagonal plate (green), and Koch snowflake (orange) along with BC-snow external (dashed lines)
and internal (solid lines) are shown. (d) Observations (black) are obtained from open-field
experiments in New York (Brandt et al., 2011). BC is assumed to be externally mixed with snow
spheres (blue), spheroids (red), hexagonal plates (green), and Koch snowflakes (orange). The
effective radii ($R_e$) for each snow shape are obtained to best match observations at wavelengths of
1–1.3 μm. (e) Observations (circles) are obtained from laboratory measurements (Hadley and
Kirchstetter, 2012). BC is assumed to be externally (solid lines) and internally (dashed lines) mixed
with snow spheres.






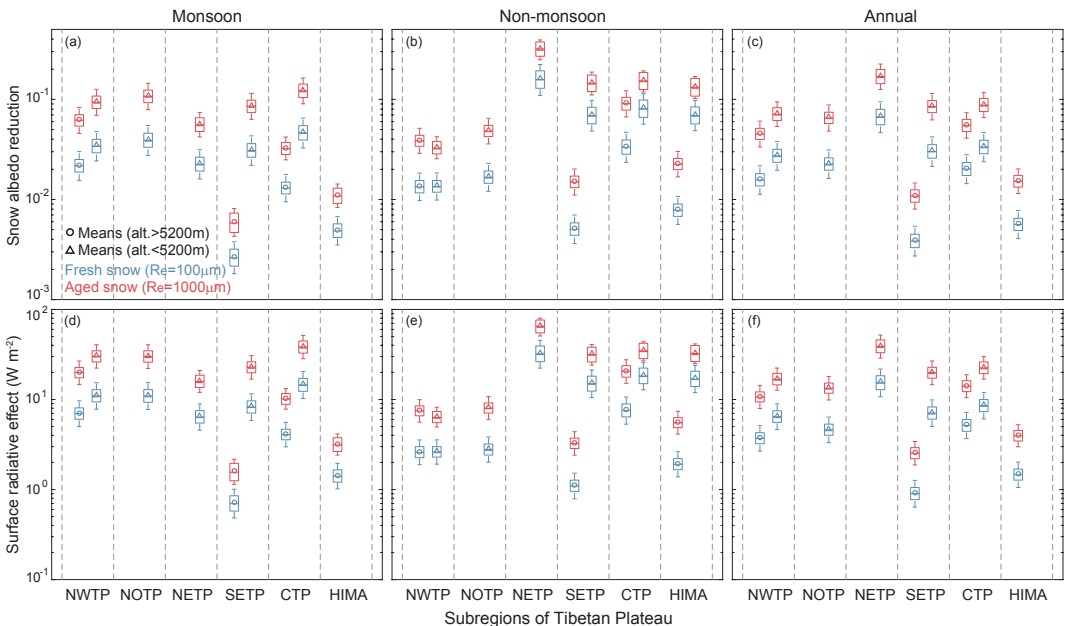

**Figure 8.** Regional and seasonal mean BC-induced all-sky snow albedo reductions and surface radiative effects during (a, d) monsoon, (b, e) non-monsoon, and (c, f) annual periods in 2000–2015 over six Tibetan Plateau (TP) sub-regions (see Fig. 1), including northwestern TP (NWTP), north of TP (NOTP), northeastern TP (NETP), southeastern TP (SETP), central TP (CTP), and Himalayas (HIMA). (a–c) Boxplots of mean snow albedo reductions within each sub-region based on SNICAR simulations using the observed BC concentrations in snow (Fig. 1), snow thicknesses, and snow densities (see text for details). Results for altitudes >5200 m and <5200 m are shown as left and right boxplots within each sub-region, respectively, with circles and triangles indicating mean values. Model results assume BC externally and internally mixed with spheres, spheroids, hexagonal plates, and Koch snowflakes for fresh (blue, $R_e$ = 100 μm) and aged (red, $R_e$ = 1000 μm) snow. Each data point used for the boxplot is the sub-regional average assuming a type of snow shape and BC-snow mixing, and hence the boxplot indicates the variation caused by effects of snow shape and BC-snow mixing state. Note that some sub-regions only have BC observations at altitudes >5200 m or <5200 m (see Fig. 1). (d–f) Same as (a–c), but for BC-induced all-sky surface radiative effects caused by the snow albedo reductions shown in (a–c). Calculations use the surface downward solar radiation and cloud cover fraction from the MERRA-2 reanalysis fields (see text and Table S2 for details).