# Peer review of "Supporting Information for"

_Atmospheric Chemistry and Physics, 2018_

## Referee Comment (RC1) · Anonymous Referee #1 · 18 Jun 2018

General Comments:

The manuscript investigates the effects of snow grain shape and BC-snow mixing sates on the snow albedo and surface radiative forcing over the Tibetan Plateau. To achieve the goal, the authors improve the SNICAR model parameterization by introducing non-spherical snow grain shape and BC-snow mixing states based on their previous work, and the parameterization is systematically compared with observations of both pure and polluted snow. Furthermore, the BC observation in the TP is well reviewed, and the uncertainties related to the snow shape and BC-snow mixing are studied. The
topic is interesting and important for snow albedo studies, and the manuscript is well organized and written. It can be published on ACP after minor revision.

Specific Comments:

Title: The title of the manuscript is not very clear, and the main focus of the paper cannot be clearly obtained through the title. The snow grain shape effects are not related to the BC.

Line 278-287: There are significant uncertainties on BC MAC. The difference between He et al. (2017b) and Bond and Bergstrom (2006) can be simply explained by natural variations. However, the authors made unrealistic adjustment on BC density and size. Is this really necessary, and how would a different MAC in the model influence the final results?

Table 1: The authors made some assumptions to evaluate the new parameterization, and Table 1 list most parameters for comparison with observations. The detailed assumptions should be indicated in the manuscript, e.g., which parameters are assumed, and which parameters are observed. Meanwhile, are the parameters adjusted to match the observations, or realistic parameters that are picked independent of observations lead to the great agreement.

Figure 6: It seems that most observations give an albedo slightly less than 1 around 400nm, whereas most model results overestimate the albedo. Is there any explanation?

Figure 8: The effects on the snow albedo and surface radiative effects are illustrated in the figure. The two variables are closely related, and, from the figure, it seems that there is a strong correlation between them.

The manuscripts show significant influences of snow shape and BC-snow mixing on surface albedo. During the discussion, the albedo reductions, which are relatively small, are used to evaluate the influence. The surface albedos under different circumstances can directly compared to indicate the influences. Furthermore, considering the variations on the models and input parameters, the uncertainties on the albedo may be quite significant, and this may greatly influence the conclusions.

The manuscript includes a lot of information and leads to a few quite important conclusions. The conclusion section seems simply a list of the work done and conclusions obtained. A lot of details are included in the section, but it is not well organized. It should definitely be re-organized to better summary the focus of the manuscript.

---

## Referee Comment (RC2) · Anonymous Referee #2 · 29 Jun 2018

In this paper, the authors study the impact of snow grain shape and black carbon (BC)-in-snow mixing state on snow albedo and BC-snow radiative effects. The authors update the SNICAR model by introducing new sets of parameterizations for snow optical properties based on snow grain shape and BC-in-snow mixing state. The updated SNICAR model is used to reproduce spectral observations of pure and BC-contaminated snow, and is applied to field observations across Tibetan Plateau to illustrate the impact of snow grain shape and BC-in-snow mixing state on regional BC-snow radiative effects. The discussions and figures are clear and well organized in

general.

Specific comments:

1. Table 1: For field observations that did not measure underlying ground albedo, the authors assume an albedo of 0 for SNICAR computation; while the underlying ground albedo rarely reach 0 even for dark soil. The snow depth for some of these measurements is shallow, that some light may penetrate through the snowpack. Is there any reason that the authors assigned 0? Perhaps consider adjusting underlying ground albedo to see if this will impact the comparisons show in Figure 6 and 7.

2. Table 1: For field studies that report snow effective radius, how did they define/measure/derive the snow effective radius? Do they use similar assumptions as the spherical snow grain in SNICAR?

3. Lines 325-326: the authors say they "... made reasonable assumptions for cases when measurements are absent". The readers may wonder what are these "reasonable assumptions" and how did authors justified these assumptions. Perhaps including some details on, for example, how to assign underlying ground albedo (comment 1) when measurement is absent, and etc.

4. Table 2: The zeros in albedo reduction values can be distracting that prevent direct comparisons across regions; perhaps consider keeping only the non-zero digits and modify the unit.

5. Lines 340-342: in Figure 6a, as the authors mentioned, the snow grains created in Hadley and Kirchstetter (2012) tend to be spherical, yet the nonspherical grain assumption yields better results. What does this imply for future modeling/field works regarding snow grain shape and snow grain size? Does this mean even if the snow grain shape is relatively well observed in the field, the snow radiative transfer modeling based off the observed grain shape may not improve the snow modeling? Or in another word, to what extend should radiative modeling rely on field observed snow grain

shapes since it seems, from figure 6b, the model can always adjust snow grain size to match observations, no matter what grain shape it adopted.

6. Figure 6b and 7d: it seems that the model simulations fail to capture the drop of snow albedo around 0.25 um observed by Brandt et al., 2011. Is there any explanation?

---

## Referee Comment (RC3) · Anonymous Referee #3 · 8 Jul 2018

The authors implement a set of new parameterizations in the widely used SNICAR model to account for effects of snow grain shape and the mixing state of BC-snow. Then, they apply the updated SNICAR model with in-situ measurements of BC concentrations in the Tibetan Plateau snowpack to quantify the present-day snow albedo effects. Generally, the results are of great significance, and it's a very interesting paper with well written, and the expression is clear. I suggest that this manuscript could be accepted with minor revisions. Minor comments to author: 1) My major concern is that the historical snow sampling sites are very limited in the TP regions, and some of the

sampling sites are only representative the high glacier regions. The author should be very careful to use the surface measurement to represent the regional averages. So I don't think it is quite useful to divide the entire TP and surrounding areas into six sub-regions as shown in Figure 1 and Table 2. 2) The conclusion is a little repetitive, which should be reconstructed.

---

## Author Response (AR1)

**A Point-by-Point Response to Review Comments**

Dr. Yan Yin
Editor, *Atmospheric Chemistry and Physics*

Dear Dr. Yin,

We are submitting a revised manuscript (#acp-2018-476) for your consideration of publication in *Atmospheric Chemistry and Physics*. We have carefully studied the reviewers' comments and carried out revisions accordingly. Below is a point-by-point response (marked as red) to the review comments. We have also provided a copy of track-change manuscript as well as a clean copy of the revised manuscript.

Thank you for your consideration of this submission. We hope you find our responses adequately address the review comments and the revisions acceptable. We would greatly appreciate it if you could get back to us with your decision at your earliest convenience.

Sincerely,

Cenlin He
National Center for Atmospheric Research
Boulder, CO 80301, USA

**Referee #1**

"The manuscript investigates the effects of snow grain shape and BC-snow mixing sates on the snow albedo and surface radiative forcing over the Tibetan Plateau. To achieve the goal, the authors improve the SNICAR model parameterization by introducing nonspherical snow grain shape and BC-snow mixing states based on their previous work, and the parameterization is systematically compared with observations of both pure and polluted snow. Furthermore, the BC observation in the TP is well reviewed, and the uncertainties related to the snow shape and BC-snow mixing are studied. The topic is interesting and important for snow albedo studies, and the manuscript is well organized and written. It can be published on ACP after minor revision."

We thank the reviewer for his/her constructive comments and suggestions, which help to improve the manuscript. Below is a point-by-point response to the comments.

**Specific Comments:**
1. Title: The title of the manuscript is not very clear, and the main focus of the paper cannot be clearly obtained through the title. The snow grain shape effects are not related to the BC.

Response: Thank you for the comments. First, we would like to clarify that the snow grain shape effects are closely related to BC impacts on snow albedo. As we showed in this work (and our previous study, He et al. 2018a JGR), spherical snow grains lead to stronger BC-induced albedo reductions than nonspherical snow grains if other conditions/variables are the same. Both snow shape and aerosol-snow mixing state are important to BC-snow albedo effects. In fact, one of our highlights in this work is that the combination of snow grain shape and BC-snow mixing state shows an important interactive effect on BC-induced albedo reduction. Second, the focus of this paper is to assess the uncertainty in BC-induced snow albedo reduction over the Tibetan Plateau caused by snow grain shape and BC-snow mixing state using an improved SNICAR model, which is consistent with the current title. Thus, we think the current title can reflect the focus of the paper and we choose not to change it. Please note that we also put some efforts in describing and evaluating the implementation of new aerosol-snow parameterizations into SNICAR in this paper, because this is the modeling basis for quantifying snow albedo uncertainties over the Tibetan Plateau, which does not deviate from the paper focus.

2. Line 278-287: There are significant uncertainties on BC MAC. The difference between He et al. (2017b) and Bond and Bergstrom (2006) can be simply explained by natural variations. However, the authors made unrealistic adjustment on BC density and size. Is this really necessary, and how would a different MAC in the model influence the final results?

Response: Thank you for the comments.

First, we agree that the differences in BC MAC between He et al. (2017b) and Bond and Bergstrom (2006) could be due to natural variations/uncertainties. In fact, BC MAC could vary from ~2 to ~15 $m^2$ $g^{-1}$ due to uncertainties in particle density, size, structure, and refractive index. However, based on a comprehensive review of observations, Bond and Bergstrom (2006) recommended a value of 7.5 $m^2$ $g^{-1}$ at 550 nm to best represent BC MAC, which has been widely adopted in previous studies (e.g., Aoki et al., 2011; Flanner et al., 2007, 2009). Thus, to reduce the potential uncertainty from BC MAC in this work, we have chosen to use the value recommended by Bond and Bergstrom (2006).

Second, to achieve the recommended BC MAC, we adjusted the BC density to be 1.5 g $cm^{-3}$ and BC size to be a lognormal distribution with a geometric mean diameter of 0.06 μm and a geometric standard deviation of 1.5. We would like to clarify that these values are reasonable for BC particles. (1) In fact, a BC density of 1.5 g $cm^{-3}$ has been widely used in previous studies (e.g., Flanner et al. 2007; Aoki et al., 2011), as indicated in the manuscript. Bond and Bergstrom (2006) suggested that the measured void-free BC usually has a density of 1.7–1.9 g $cm^{-3}$ but the density can be lower for BC with voids. Long et al. (2013) further showed that ambient BC particle density can vary from 1.2 to 1.8 g $cm^{-3}$. (2) The BC size used in this work is also within the observed ranges. Bond et al. (2006) showed that the observed BC geometric mean diameter varies from 0.01 to 0.15 μm near combustion sources, while the observed geometric standard deviation varies from 1.2 to 2.0 for BC either near combustion sources or in continental plumes.

Third, if using a smaller BC MAC (e.g., 6.8 $m^2$ $g^{-1}$ at 550 nm as used in He et al. 2017b), the BC-induced snow albedo reduction would be smaller, compared with current estimates using a value of 7.5 $m^2$ $g^{-1}$. The quantification of MAC effects on snow albedo reduction is beyond the scope of this study and will be investigated in future work.

To clarify, we have included the aforementioned discussions in the track-change manuscript (Lines 292–302) as follows:

*"We should note that BC MAC could vary significantly in reality (e.g., from 2 to 15 $m^2$ $g^{-1}$ at 550 nm) due to uncertainties from particle density, size, structure, and refractive index (Bond and Bergstrom, 2006). Thus, we use the recommended value (7.5 $m^2$ $g^{-1}$) derived from a comprehensive review of measurements to reduce the potential uncertainty from BC MAC in this study. Compared with the current estimates, using a smaller BC MAC (e.g., 6.8 $m^2$ $g^{-1}$ at 550 nm as used in He et al. 2017b) would lead to weaker BC-induced snow albedo reductions, the quantification of which, however, is beyond the scope of this study and will be investigated in future work. In addition, the adjusted BC density and size used in the present study are still within the observed ranges, with 1.2–1.9 g $cm^{-3}$ for densities (Bond and Bergstrom, 2006; Long et al., 2013) as well as 0.01–0.15*

*µm and 1.2–2.0 for geometric mean diameters and standard deviations (Bond et al., 2006), respectively."*

3. Table 1: The authors made some assumptions to evaluate the new parameterization, and Table 1 list most parameters for comparison with observations. The detailed assumptions should be indicated in the manuscript, e.g., which parameters are assumed, and which parameters are observed. Meanwhile, are the parameters adjusted to match the observations, or realistic parameters that are picked independent of observations lead to the great agreement.

Response: Thank you for the comments. We would like to clarify that all the parameter values are picked based on the corresponding observed/realistic values in each case when the observations are available. We did not adjust model parameters to match observations. Even for the assumed parameter values indicated in Table 1, we did not tune the values to match observations. Instead, we adopted either commonly used values or observed values from other studies. Following the reviewer's comment, we have included the detailed assumptions and clarifications in the track-change manuscript as follows:

Lines 332-346: *"To conduct reasonable comparisons, we used the observed snow density, grain size, thickness, snowpack layer, direct/diffuse radiation, solar zenith angle, and underlying ground albedo in model simulations for each case (see Table 1 and Figure 6 for details), except for underlying ground albedos in the Brandt et al. (2011) and Painter et al. (2007) cases and the grain size of the second snow layer in the Brandt et al. (2011) case because of unavailable measurements. Thus, we assumed black underlying grounds (albedo = 0) in the two cases, which has negligible effects on albedo estimates due to thick snow optical depths. In the Brandt et al. (2011) case, we further assumed an effective radius of 500 µm (typical for aged snow) in the second snow layer to make it optically semi-infinite, which is consistent with the observed condition."*

Lines 386-395: *"Similar to Section 3.4.1, we used the observed BC concentration in snow, snow density, grain size, thickness, snowpack layer, direct/diffuse radiation, solar zenith angle, and underlying ground albedo in model simulations for each case (see Table 1 and Figure 7 for details), except for the snow density in the Pedersen et al. (2015) case and the underlying ground albedo in the Meinander et al. (2013) case because of unavailable measurements. Thus, we assumed a typical fresh snow density of 150 kg m$^{-3}$ in the former case and a black underlying ground (albedo = 0) in the latter case. Compared with assuming a black underlying ground, we find that using a non-black underlying ground albedo typically observed over the Tibet (Qu et al., 2014) only leads to very small (<5%) relative differences in albedo calculations in the Meinander et al. (2013) case."*

4. Figure 6: It seems that most observations give an albedo slightly less than 1 around 400nm, whereas most model results overestimate the albedo. Is there any explanation?

Response: Thanks for pointing it out. The slight but systematic model overestimates at around 400 nm (shown in Fig. 6) are probably due to the uncertainty of ice refractive indices. Based on a recent study (Picard et al., 2016), the ice refractive indices (Warren and Brandt, 2008) used in this study may result in too weak snow absorption around 400 nm and hence lead to albedo overestimates, compared with observations. We have included the following discussions in the track-change manuscript (Lines 368–380):

*"We note that model results in all cases show slight but consistent albedo overestimates around 400 nm compared with observations (Fig. 6), probably due to the uncertainty of ice refractive indices. In this work, we used ice refractive indices from the most widely-used database (Warren and Brandt, 2008) obtained from measurements in the Antarctic, which shows a very low ice absorption coefficient around 400 nm. However, based on more recent measurements in Antarctic snow, Picard et al. (2016) found a much higher ice absorption coefficient around 400 nm than that from Warren and Brandt (2008), which suggested that the uncertainty in ice visible absorption is probably larger than generally appreciated. Therefore, the weak snow absorption caused by refractive indices used in this study could lead to the overestimates in modeled albedo around 400 nm."*

5. Figure 8: The effects on the snow albedo and surface radiative effects are illustrated in the figure. The two variables are closely related, and, from the figure, it seems that there is a strong correlation between them.

Response: Yes, the BC-induced snow albedo reduction is closely correlated with the surface radiative effects. This is because the regional mean surface radiative effect is computed by multiplying the regional mean snow albedo reductions with the regional mean surface downward solar fluxes (from MERRA-2 reanalysis data). As shown in Table S2 (in the supplement), the mean surface downward solar fluxes across different Tibetan sub-regions are similar during the same season, which leads to the strong correlation between snow albedo reductions and surface radiative effects across the sub-regions as shown in Fig. 8.

6. The manuscripts show significant influences of snow shape and BC-snow mixing on surface albedo. During the discussion, the albedo reductions, which are relatively small, are used to evaluate the influence. The surface albedos under different circumstances can directly compared to indicate the influences. Furthermore, considering the variations on the models and input parameters, the uncertainties on the albedo may be quite significant, and this may greatly influence the conclusions.

Response: Thank you for the comments. We agree that the present estimates of albedo reductions may be associated with uncertainties from various factors, including model and input parameters, which could affect the signal of BC-induced albedo reductions. Besides, in relatively clean areas, the BC-induced albedo reductions are small (e.g.,

<0.01), which may be comparable or even smaller than the uncertainty of surface/snow albedo under different conditions. Surface albedos obtained from remote sensing observations typically have errors of a few percent (*Warren*, 2013 JGR). However, in the polluted regions, the albedo reductions can be larger than 0.1, which provides strong and detectable signals. In this study, to reduce the uncertainty in albedo calculations, we have used observed values for model/input parameters in the estimates of BC-induced albedo reductions over TP when measurements are available. However, we do realize that there are still several important uncertainty sources and limitations in this study, including uncertainties from measurements, BC and snow grain properties, and complex snowpack processes, which have been discussed in the original manuscript (Lines 470–482). Here, we have further included discussions on the uncertainty issues mentioned by the reviewer in the track-change manuscript (Lines 533–536) as follows:

*"These uncertainties associated with modeling and measurements may decrease the signal-to-noise ratio for the detection of BC effects on snow albedo, particularly in relatively clean regions with small BC-induced albedo reductions (e.g., <0.01). Thus, improved and robust estimates require both accurate snow albedo modeling and snowpack measurements."*

7. The manuscript includes a lot of information and leads to a few quite important conclusions. The conclusion section seems simply a list of the work done and conclusions obtained. A lot of details are included in the section, but it is not well organized. It should definitely be re-organized to better summary the focus of the manuscript.

Response: Thank you for the comments. We have re-organized and refined the conclusion section to better summarize and highlight the focus of this study as follows (Lines 544–617):

" *We implemented a set of new BC-snow parameterizations into SNICAR, a widely used snow albedo model, to account for the effects of snow nonsphericity and BC-snow internal mixing. We evaluated model simulations by comparing with observations. We further applied the updated SNICAR model with a comprehensive set of in-situ measurements of BC concentrations in the Tibetan Plateau (TP) snowpack (glacier) to quantify the present-day BC-induced snow albedo effects and associated uncertainties from snow grain shape and BC-snow mixing state.*

*Based on the SNICAR model updated with new BC-snow parameterizations, we found that nonspherical snow grains tend to have higher pure albedos but lower BC-induced albedo reductions compared with spherical snow grains, while BC-snow internal mixing substantially enhances albedo reductions relative to external mixing. Compared with observations, model simulations assuming nonspherical snow grains and BC-snow internal mixing perform better than those with the common assumption of snow spheres and external mixing. The results suggest an important interactive effect from snow*

*nonsphericity and internal mixing, and highlight the necessity of concurrently accounting for the two factors in snow albedo and climate modeling.*

*We further applied the updated SNICAR model with comprehensive in-situ observations of BC concentrations in snow and snowpack properties over the TP to quantify the present-day (2000–2015) BC-induced snow albedo effects. We found that BC concentrations show distinct sub-regional and seasonal variations. The concentrations are generally higher in the non-monsoon season and low-altitudes (<5200 m) than in the monsoon season and high-altitudes (>5200 m), respectively. The spatiotemporal distributions of snow albedo reductions and surface radiative effects generally follow that of BC concentrations. As a result, the BC-induced mean albedo effects vary by up to an order of magnitude across different sub-regions and seasons, with values of 0.7–30.7 (1.4–58.4) W $m^{-2}$ for BC externally mixed with fresh (aged) snow spheres.*

*Moreover, the BC-snow albedo effects over the TP are significantly affected by the uncertainty in snow grain shape and BC-snow mixing state. We found that BC-snow internal mixing enhances the mean albedo effects by 30–60% relative to external mixing across different sub-regions and seasons, while nonspherical snow grains reduce the albedo effects by up to 31% relative to spherical grains. These effects become comparably important with the snow aging/size effect over polluted areas. Therefore, the combined effects of snow grain shape and BC-snow mixing state can complicate the spatiotemporal features of BC-snow albedo effects over the TP, with significant implications for regional hydrological processes and water management.*

*In summary, this study points toward an imperative need for improved measurements and model characterization of snow grain shape and aerosol-snow mixing state in order to accurately estimate BC-induced snow albedo effects. In future work, we will incorporate the new features of the updated SNICAR model into land surface and climate models, including CESM-Community Land Model (CLM) for global modeling and WRF-Noah-MP for regional modeling, to account for the effects of snow grain shape and aerosol-snow mixing state and to assess the associated uncertainties and hydrological feedbacks in global/regional climate system.”*

**Referee #2**

"In this paper, the authors study the impact of snow grain shape and black carbon (BC)-in-snow mixing state on snow albedo and BC-snow radiative effects. The authors update the SNICAR model by introducing new sets of parameterizations for snow optical properties based on snow grain shape and BC-in-snow mixing state. The updated SNICAR model is used to reproduce spectral observations of pure and BC contaminated snow, and is applied to field observations across Tibetan Plateau to illustrate the impact of snow grain shape and BC-in-snow mixing state on regional BC-snow radiative effects. The discussions and figures are clear and well organized in general."

We thank the reviewer for his/her constructive comments and suggestions, which help to improve the manuscript. Below is a point-by-point response to the comments.

**Specific Comments:**
1. Table 1: For field observations that did not measure underlying ground albedo, the authors assume an albedo of 0 for SNICAR computation; while the underlying ground albedo rarely reach 0 even for dark soil. The snow depth for some of these measurements is shallow, that some light may penetrate through the snowpack. Is there any reason that the authors assigned 0? Perhaps consider adjusting underlying ground albedo to see if this will impact the comparisons show in Figure 6 and 7.

Response: Thank you for the comments. We have conducted additional sensitivity simulations for the three cases without measurements of underlying ground albedo by using values of 0.1 and 0.2 for visible and NIR bands, respectively, based on observations over the TP. For the Painter et al. (2007) and Brandt et al. (2011) cases, the differences by using different underlying ground albedos are negligible due to thick snow optical depths. For the Meinander et al. (2013) case with relative thin snow layers, the relative differences by using different underlying ground albedos are still small (<5%) due to large snow grain sizes and hence thick optical depths. We have included these discussions in the track-change manuscript as follows:
   Lines 337–338: "*Thus, we assumed black underlying ground (albedo = 0) in the two cases, which has negligible effects on albedo estimates due to thick snow optical depths.*"
   Lines 392–395: "*Compared with assuming a black underlying ground, we find that using a non-black underlying ground albedo typically observed over the Tibet (Qu et al., 2014) only leads to very small (<5%) relative differences in albedo calculations in the Meinander et al. (2013) case.*"

2. Table 1: For field studies that report snow effective radius, how did they define/measure/derive the snow effective radius? Do they use similar assumptions as the spherical snow grain in SNICAR?

Response: Thank you for the comments. Snow grain sizes reported by the field studies are retrieved by different methods. For the Painter et al. (2007), Hadley and Kirchstetter (2012), and Pedersen et al. (2015) cases, they retrieved snow grain sizes by matching results of snow radiative transfer models with measured NIR snow albedo spectra. For the Grenfell et al. (1994), Meinander et al. (2013), and Svensson et al. (2016) cases, they determined snow grains sizes by visual estimates with tools (e.g., hand lens with macro-photograph or mm-grids with magnifier). We note that these different measuring methods could introduce uncertainties to the measured snow grain sizes. Moreover, the snow grain size from visual estimates in field studies also differs from the snow effective size (i.e., surface area-weighted mean radius) defined in SNICAR, which could introduce additional uncertainties to snow albedo calculations and comparisons with observations. We have included these discussions in the revised manuscript (Lines 428–434) as follows:

*"We note that the snow grain sizes reported by the aforementioned field studies are retrieved by different methods, including matching snow model results with measured albedo spectra (Painter et al., 2007; Hadley and Kirchstetter, 2012; Pedersen et al., 2015) and visual estimates with tools (Grenfell et al., 1994; Meinander et al., 2013; Svensson et al., 2016) that are not equivalent to the snow effective size (i.e., surface area-weighted mean radius) defined in SNICAR. This could introduce uncertainties to snow albedo calculations and model-observation comparisons."*

3. Lines 325-326: the authors say they "made reasonable assumptions for cases when measurements are absent". The readers may wonder what are these "reasonable assumptions" and how did authors justify these assumptions. Perhaps including some details on, for example, how to assign underlying ground albedo (comment 1) when measurement is absent, and etc.

Response: Thank you for the comments. Following the reviewer's suggestion, we have included more details, including which parameters are based on observed values and which parameters are assumed as well as some justifications of these assumptions. We have also conducted sensitivity simulations to investigate effects of assumed underlying ground albedo (see the response to Comment #1). The additional discussions have been included in the track-changed manuscript (Lines 332–346 and 386–395). Please see the response to Reviewer #1, Comment #3 for details.

4. Table 2: The zeros in albedo reduction values can be distracting that prevent direct comparisons across regions; perhaps consider keeping only the non-zero digits and modify the unit.

Response: Thanks for the comment. Following the reviewer's suggestion, we have modified the values using the expression of scientific notation in the revised Table 2.

5. Lines 340-342: in Figure 6a, as the authors mentioned, the snow grains created in Hadley and Kirchstetter (2012) tend to be spherical, yet the nonspherical grain assumption yields better results. What does this imply for future modeling/field works regarding snow grain shape and snow grain size? Does this mean even if the snow grain shape is relatively well observed in the field, the snow radiative transfer modeling based off the observed grain shape may not improve the snow modeling? Or in another word, to what extend should radiative modeling rely on field observed snow grain shapes since it seems, from figure 6b, the model can always adjust snow grain size to match observations, no matter what grain shape it adopted.

Response: Thanks for the comments.

(1) The snow grains created in Hadley and Kirchstetter (2012) tend to be spherical. However, based on their microscopic images (Fig. S3 in their study), the grains are not perfectly spherical and there are still a portion of grains with either spheroid or aggregating shapes. This is probably why assuming nonspherical grains in our model yields slightly better results than assuming purely spherical grains (Fig. 6a in this study).

(2) Our results do not imply that the snow albedo modeling based on observed grain shape may not improve model results. Instead, one of our key findings/points in this study is that it is necessary to account for realistic/observed snow grain shape in order to accurately estimate snow albedo, which has been supported by the improved model results using nonspherical snow grains (see Sect. 3.4). However, each parameter used in snow modeling could be associated with uncertainties. It is likely that using the observed grain shape may not improve model results when the uncertainties/biases in other model parameters are large. Thus, accurate estimates of snow albedo require constraining all the model parameters together by observations. In summary, this study points toward an imperative need for improved measurements and model characterization of snow shapes.

(3) We agree that the snow grain size can always be adjusted to make model results match observations, whatever the grain shape is assumed. However, this could lead to the issue of getting right answers (e.g., albedo) for wrong reasons (e.g., grain size) due to the lack of grain shape information. Moreover, we have shown that assuming different snow grain shapes can lead to substantial variations in the optimal grain sizes determined by matching observed albedo spectra (Figs. 6b and 7d). This highlights the necessity of accounting for realistic grain shapes in snow size retrievals, which can effectively reduce the uncertainty in retrieved grain sizes. In addition, for the purpose of weather and climate forecasts, snow and climate modeling rely on observed/realistic snow grain shape to produce accurate albedo predictions and associated climate feedbacks.

6. Figure 6b and 7d: it seems that the model simulations fail to capture the drop of snow albedo around 0.25 um observed by Brandt et al., 2011. Is there any explanation?

Response: Thanks for pointing it out. This is because of the uncertainty in ice refractive indices at short wavelengths. The ice refractive indices used in this study result in too weak snow absorption at wavelengths <400 nm and hence lead to model overestimates in albedo at these wavelengths. We have included the discussions in the track-change manuscript (Lines 368–380). Please see the response to Reviewer #1, Comment #4 for details.

**Referee #3**

"The authors implement a set of new parameterizations in the widely used SNICAR model to account for effects of snow grain shape and the mixing state of BC-snow. Then, they apply the updated SNICAR model with in-situ measurements of BC concentrations in the Tibetan Plateau snowpack to quantify the present-day snow albedo effects. Generally, the results are of great significance, and it's a very interesting paper with well written, and the expression is clear. I suggest that this manuscript could be accepted with minor revisions."

We thank the reviewer for his/her constructive comments and suggestions, which help to improve the manuscript. Below is a point-by-point response to the comments.

**Minor Comments:**
1. My major concern is that the historical snow sampling sites are very limited in the TP regions, and some of the sampling sites are only representative the high glacier regions. The author should be very careful to use the surface measurement to represent the regional averages. So I don't think it is quite useful to divide the entire TP and surrounding areas into six subregions as shown in Figure 1 and Table 2.

Response: Thanks for the comments. We agree that the snow sampling sites are very limited over the TP and may not be representative for the entire TP region. This is why we have divided the entire TP domain into six smaller sub-regions for analysis. Within each sub-region, we found that BC concentrations show distinct altitudinal and seasonal variations. Thus, we have further divided each sub-region into high-/low-altitude areas and monsoon/non-monsoon seasons for analysis. As such, we tried to reduce the uncertainty from sample representativeness. Therefore, we believe that dividing the entire TP into smaller sub-regions is still useful. However, we do realize that even after dividing the sub-regions, the current observations in each sub-region are still limited, which may introduce uncertainty in the analysis and highlights an urgent need for more extensive measurements in the region. We have included these discussions in the manuscript (Lines 181–185) as follows:

*"We note that the current observations over the TP are still rather limited spatially and temporally, leading to questions of representativeness and introducing uncertainty in the analysis. Thus, the large sub-regional, altitudinal, and seasonal heterogeneity of BC concentrations in the TP snowpack highlights an urgent need for extensive measurements."*

2. The conclusion is a little repetitive, which should be reconstructed.

Response: Thanks for the comments. We have re-organized and refined the conclusion section to better summarize and highlight the focus of this study (Lines 544–617). Please see the response to Reviewer #1, Comment #7 for details.

[revised manuscript text omitted]
and Bergstrom, 2006). Thus, we use the recommended value (7.5 m$^2$ g$^{-1}$) derived from a
comprehensive review of measurements to reduce the potential uncertainty from BC MAC in this
study. Compared with the current estimates, using a smaller BC MAC (e.g., 6.8 m$^2$ g$^{-1}$ at 550 nm
as used in He et al. 2017b) would lead to weaker BC-induced snow albedo reductions, the
quantification of which, however, is beyond the scope of this study and will be investigated in
future work. In addition, the adjusted BC density and size used in the present study are still within
the observed ranges, with 1.2–1.9 g cm$^{-3}$ for densities (Bond and Bergstrom, 2006; Long et al.,
2013) as well as 0.01–0.15 μm and 1.2–2.0 for geometric mean diameters and standard deviations
(Bond et al., 2006), respectively.

[revised manuscript text omitted]

We note that model results in all cases show slight but consistent albedo overestimates around 400 nm compared with observations (Fig. 6), probably due to the uncertainty of ice refractive indices. In this work, we used ice refractive indices from the most widely-used database (Warren and Brandt, 2008) obtained from measurements in the Antarctic, which shows a very low ice absorption coefficient around 400 nm. However, based on more recent measurements in

Antarctic snow, Picard et al. (2016) found a much higher ice absorption coefficient around 400 nm than that from Warren and Brandt (2008), which suggested that the uncertainty in ice visible absorption is probably larger than generally appreciated. Therefore, the weak snow absorption caused by refractive indices used in this study could lead to the overestimates in modeled albedo
around 400 nm.

**3.4.2 BC-contaminated snow albedo**

We further compared BC-contaminated snow albedo from SNICAR simulations with
observations (Fig. 7) from laboratory measurements (Hadley and Kirchstetter, 2012), open-field
experiments (Brandt et al., 2011; Svensson et al., 2016), and field measurements in the Arctic
(Meinander et al., 2013; Pedersen et al., 2015). Similar to Section 3.4.1, we used the observed BC
concentration in snow, snow density, grain size, thickness, snowpack layer, direct/diffuse radiation,
solar zenith angle, and underlying ground albedo in model simulations for each case (see Table 1
and Figure 7 for details), except for the snow density in the Pedersen et al. (2015) case and the
underlying ground albedo in the Meinander et al. (2013) case because of unavailable
measurements. Thus, we assumed a typical fresh snow density of 150 $kg\ m^{-3}$ in the former case
and a black underlying ground (albedo = 0) in the latter case. Compared with assuming a black
underlying ground, we find that using a non-black underlying ground albedo typically observed
over the Tibet (Qu et al., 2014) only leads to very small (<5%) relative differences in albedo
calculations in the Meinander et al. (2013) case. Due to the lack of measurements, we further
assumed BC internally or externally mixed with different snow shapes in the simulations to
quantify the combined effects of BC-snow mixing state and snow grain shape.

Compared with the widely-used assumption of BC externally mixed with spherical snow
grains, we find that accounting for both internal mixing and snow nonsphericity improves model
simulations (Fig. 7). For example, assuming BC-sphere external mixing leads to a systematical
underestimate of polluted snow albedo for <2000 ppb BC compared with the observations from
Svensson et al. (2016), while assuming BC-Koch snowflake internal mixing reduces the model
underestimate (Fig. 7b), with the normalized mean bias (NMB) and root-mean-square error
(RMSE) decreasing from -0.04 to 0.01 and from 0.033 to 0.019, respectively. Similarly, in the
observational case of Pedersen et al. (2015), simulations assuming BC-spheroid external mixing
perform better than those assuming BC-sphere external mixing (Fig. 7a), reducing the NMB from
-0.012 to -0.003 and RMSE from 0.028 to 0.025. Compared with the observations of Meinander
et al. (2013), model results using spherical snow grains underestimate the spectral snow albedo
contaminated by BC (Fig. 7c), regardless of model assumptions of BC-snow mixing state. Using nonspherical grains instead increases the simulated albedo and reduces model biases in this case, although it is still unable to fully capture the observed pattern (Fig. 7c). Considering that snow grains tend to be spherical in the observations from Hadley and Kirchstetter (2012), we assumed

BC-sphere external/internal mixing in the comparisons. The model results with external mixing are systematically biased high, particularly for large BC concentrations (>110 ppb), while using internal mixing effectively reduces the albedo overestimates (Fig. 7e). As such, the observations fall between the results of external and internal mixing, suggesting a combination of partial external and internal mixing would best match the observations. Compared with the way of increasing BC MAC for BC-snow external mixing to reduce model overestimates in polluted snow albedo, which was used in Hadley and Kirchstetter (2012), the present study provides a physically- based alternative (i.e., internal mixing) for model improvements. In fact, it is very likely that a large portion of BC is internally mixed with snow grains in the experiments of Hadley and

Kirchstetter (2012), since they produced the BC-contaminated snow via freezing of aqueous hydrophilic BC suspensions.

We note that the snow grain sizes reported by the aforementioned field studies are retrieved by different methods, including matching snow model results with measured albedo spectra (Painter et al., 2007; Hadley and Kirchstetter, 2012; Pedersen et al., 2015) and visual estimates with tools (Grenfell et al., 1994; Meinander et al., 2013; Svensson et al., 2016) that are not equivalent to the snow effective size (i.e., surface area-weighted mean radius) defined in SNICAR.

This could introduce uncertainties to snow albedo calculations and model-observation comparisons.

[revised manuscript text omitted]

These uncertainties associated with modeling and measurements may decrease the signal-to-noise
ratio for the detection of BC effects on snow albedo, particularly in relatively clean regions with
small BC-induced albedo reductions (e.g., <0.01). Thus, improved and robust estimates require
both accurate snow albedo modeling and snowpack measurements.

**5. Conclusions, implications, and future work**

We implemented a set of new BC-snow parameterizations into SNICAR, a widely used
snow albedo model, to account for the effects of snow nonsphericity and BC-snow internal mixing.
We evaluated model simulations by comparing with observations. We further applied the updated
SNICAR model with a comprehensive set of *in-situ* measurements of BC concentrations in the
Tibetan Plateau (TP) snowpack (glacier) to quantify the present-day BC-induced snow albedo
effects and associated uncertainties from snow grain shape and BC-snow mixing state.

Based on the SNICAR model updated with new BC-snow parameterizations, we found that
nonspherical snow grains tend to have higher pure albedos but lower BC-induced albedo
reductions compared with spherical snow grains, while BC-snow internal mixing substantially
enhances albedo reductions relative to external mixing. Compared with observations, model
simulations assuming nonspherical snow grains and BC-snow internal mixing perform better than
those with the common assumption of snow spheres and external mixing. The results suggest an
important interactive effect from snow nonsphericity and internal mixing, and highlight the
necessity of concurrently accounting for the two factors in snow albedo and climate modeling.

We further applied the updated SNICAR model with comprehensive *in-situ* observations
of BC concentrations in snow and snowpack properties over the TP to quantify the present-day
(2000–2015) BC-induced snow albedo effects. We found that BC concentrations show distinct
sub-regional and seasonal variations. The concentrations are generally higher in the non-monsoon
season and low-altitudes (<5200 m) than in the monsoon season and high-altitudes (>5200 m),
respectively. The spatiotemporal distributions of snow albedo reductions and surface radiative
effects generally follow that of BC concentrations. As a result, the BC-induced mean albedo effects
vary by up to an order of magnitude across different sub-regions and seasons, with values of 0.7–
30.7 (1.4–58.4) W m$^{-2}$ for BC externally mixed with fresh (aged) snow spheres.

Moreover, the BC-snow albedo effects over the TP are significantly affected by the
uncertainty in snow grain shape and BC-snow mixing state. We found that BC-snow internal
mixing enhances the mean albedo effects by 30–60% relative to external mixing across different
sub-regions and seasons, while nonspherical snow grains reduce the albedo effects by up to 31%
relative to spherical grains. These effects become comparably important with the snow aging/size
effect over polluted areas. Therefore, the combined effects of snow grain shape and BC-snow
mixing state can complicate the spatiotemporal features of BC-snow albedo effects over the TP,
with significant implications for regional hydrological processes and water management.

[revised manuscript text omitted]

Qin, D. H., Liu, S. Y., and Li, P. J.: Snow cover distribution, variability, and response to climate change in
western China, J. Clim., 19(9), 1820–1833, 2006.
Qu, B., Ming, J., Kang, S.-C., Zhang, G.-S., Li, Y.-W., Li, C.-D., Zhao, S.-Y., Ji, Z.-M., and Cao, J.-J.: The
decreasing albedo of the Zhadang glacier on western Nyainqentanglha and the role of light-absorbing
impurities, Atmos. Chem. Phys., 14, 11117-11128, doi:10.5194/acp-14-11117-2014, 2014.
Qu, X. and A. Hall: Assessing Snow Albedo Feedback in Simulated Climate Change. J. Climate, 19, 2617–
2630, doi:10.1175/JCLI3750.1, 2006.
Ramanathan, V., and Carmichael, G.: Global and regional climate changes due to black carbon, Nat. Geosci., 1,
221–227, doi:10.1038/Ngeo156, 2008.
Räisänen, P., Makkonen, R., Kirkevåg, A., and Debernard, J. B.: Effects of snow grain shape on climate
simulations: sensitivity tests with the Norwegian Earth System Model, The Cryosphere, 11, 2919-2942,
doi:10.5194/tc-11-2919-2017, 2017.
Schmale, J., Flanner, M., Kang, S., Sprenger, M., Zhang, Q., Guo, J., Li, Y., Schwikowski, M., and Farinotti, D.:
Modulation of snow reflectance and snowmelt from Central Asian glaciers by anthropogenic black carbon,
Sci. Rep.-UK, 7, 40501, doi:10.1038/srep40501, 2017.
Skiles, S. M. and Painter, T. H.: Daily evolution in dust and black carbon content, snow grain size, and snow
albedo during snowmelt, RockyMountains, Colorado, J. Glaciol., 63, 118–132, doi:10.1017/jog.2016.125,
2016.
Sterle, K. M., McConnell, J. R., Dozier, J., Edwards, R., and Flanner, M. G.: Retention and radiative forcing of
black carbon in eastern Sierra Nevada snow, The Cryosphere, 7, 365-374, doi:10.5194/tc-7-365-2013, 2013.
Svensson J., Virkkula A., Meinander O., Kivekäs N., Hannula H.-R., Järvinen O., Peltoniemi J.I., Gritsevich M.,
Heikkilä A., Kontu A., Neitola K., Brus D., Dagsson-Waldhauserova P., Anttila K., Vehkamäki M., Hienola
A., de Leeuw G., and Lihavainen H.: Soot-doped natural snow and its albedo–results from field experiments.
Boreal Env. Res. 21: 481–503, 2016.
Toon, O. B., McKay, C. P., Ackerman, T. P., and Santhanam, K.: Rapid calculation of radiative heating rates
and photodissociation rates in inhomogeneous multiple scattering atmospheres, J. Geophys. Res., 94, 16287–
16301, 1989.
Wang, M, B Xu, J Cao, X Tie, H Wang, R Zhang, Y Qian, PJ Rasch, S Zhao, G Wu, H Zhao, DR Joswiak, J Li,
and Y Xie: Carbonaceous Aerosols Recorded in a Southeastern Tibetan Glacier: Analysis of Temporal
Variations and Model Estimates of Sources and Radiative Forcing, Atmos. Chem. Phys., 15, 1191-1204,
doi:10.5194/acp-15-1191-2015, 2015.
Wang, X., Doherty, S. J., and Huang, J.: Black carbon and other light-absorbing impurities in snow across
northern China, J. Geophys. Res. Atmos., 118, 1471–1492, doi:10.1029/2012JD018291, 2013.
Wang, X., Pu, W., Ren, Y., Zhang, X., Zhang, X., Shi, J., Jin, H., Dai, M., and Chen, Q.: Observations and model
simulations of snow albedo reduction in seasonal snow due to insoluble light-absorbing particles during 2014
Chinese survey, Atmos. Chem. Phys., 17, 2279-2296, doi:10.5194/acp-17-2279-2017, 2017.
Warren, S. G., and R. E. Brandt; Optical constants of ice from the ultraviolet to the microwave: A revised
compilation, J. Geophys. Res., 113, D14220, doi:10.1029/2007JD009744, 2008.
Warren, S. G., and W. J. Wiscombe: A Model for the Spectral Albedo of Snow. 2. Snow Containing Atmospheric
Aerosols, J. Atmos. Sci., 37(12), 2734–2745, 1980.
Wiscombe, W. J., and Warren, S. G.: A model for the spectral albedo of snow: I. Pure snow. Journal of the
Atmospheric          Sciences,          37(12),          2712–2733,          doi:10.1175/1520-
0469(1980)037%3C2712:AMFTSA%3E2.0.CO;2, 1980.
Wu, L., Gu, Y., Jiang, J. H., Su, H., Yu, N., Zhao, C., Qian, Y., Zhao, B., Liou, K.-N., and Choi, Y.-S.: Impacts
of aerosols on seasonal precipitation and snowpack in California based on convection-permitting WRF-Chem
simulations, Atmos. Chem. Phys., 18, 5529-5547, doi:10.5194/acp-18-5529-2018, 2018.

[revised manuscript text omitted]

[*]The parameters are assumed in simulations due to the lack of measurements. Note that the assumed underlying
ground albedos have rather small effects on albedo simulations due to thick snow optical depths.

**Table 2.** Regional and seasonal mean BC-induced all-sky snow albedo reductions for fresh snow
over the Tibetan Plateau during 2000–2015. See Table S3 for results of aged snow.

[revised manuscript text omitted]